# Genetic ancestry plays a central role in population pharmacogenomics

Hsin-Chou Yang [1,2,3 ✉], Chia-Wei Chen[1], Yu-Ting Lin[1] & Shih-Kai Chu[1]

Recent studies have pointed out the essential role of genetic ancestry in population pharmacogenetics. In this study, we analyzed the whole-genome sequencing data from The 1000 Genomes Project (Phase 3) and the pharmacogenetic information from Drug Bank, PharmGKB, PharmaADME, and Biotransformation. Here we show that ancestry-informative markers are enriched in pharmacogenetic loci, suggesting that trans-ancestry differentiation must be carefully considered in population pharmacogenetics studies. Ancestry-informative pharmacogenetic loci are located in both protein-coding and non-protein-coding regions, illustrating that a whole-genome analysis is necessary for an unbiased examination over pharmacogenetic loci. Finally, those ancestry-informative pharmacogenetic loci that target multiple drugs are often a functional variant, which reflects their importance in biological functions and pathways. In summary, we develop an efficient algorithm for an ultrahigh-dimensional principal component analysis. We create genetic catalogs of ancestry-informative markers and genes. We explore pharmacogenetic patterns and establish a high-accuracy prediction panel of genetic ancestry. Moreover, we construct a genetic ancestry pharmacogenomic database Genetic Ancestry PhD (http://hcyang.stat.sinica.edu.tw/databases/genetic_ancestry_phd/).

[1] Institute of Statistical Science, Academia Sinica, Taipei, Taiwan. [2] Institute of Statistics, National Cheng Kung University, Tainan, Taiwan. [3] Institute of Public Health, National Yang-Ming University, Taipei, Taiwan. ✉email: hsinchou@stat.sinica.edu.tw

Genetic ancestry has been long recognized as a key topic in population genetics. It has been proved an influential factor that should be modeled or controlled for in -omics studies, including genetic association studies[1–3], epigenetic studies[4–10], transcriptomic studies[11–13], and proteomic studies[14–17]. With the aid of a growing number of discovered genetic variations, more and more ancestry-informative markers (AIMs) are identified to trace the origin of individuals with differential genetic backgrounds[18–31]. AIMs have been applied in various fields of studies, such as population genetics[32–36], medical genetics[37–39], forensic sciences[40–42], and animal sciences[43,44].

Trans-ancestry pharmacological effects in population pharmacogenetic[45–54] and pharmacoepigenetic studies[55–57] have received considerable attention because of their value in precision population health. This contributes to the reasons why AIMs play central roles in both pharmacodynamics (PD) and pharmacokinetics (PK). In the context of PK, AIMs are relevant to differences in drug absorption, distribution, metabolism, and excretion (ADME); downstream outcomes of PK/PD—drug response or pharmacogenomic effect (FX); adverse drug reactions (ADRs); and a drug's effective dose in different ancestry groups[52]. An example of an ancestry-informative pharmacogenetic locus (AI-PGx) is the single nucleotide polymorphism (SNP) rs1045642 on the *MDR1* gene, also known as the *ABCB1* gene. The minor allele frequency of rs1045642 differs significantly between continent ancestry groups and is associated with ADR to amitriptyline and nortriptyline and a specific drug response to morphine[51]. In the data produced by The International HapMap Project[58], the frequency of allele *A* was 0.542 in the Center d'Etude du Polymorphism Humain Utah collection; 0.489 among Japanese in Tokyo, Japan; 0.400 among Han Chinese in Beijing, China; 0.108 among Yoruba in Ibadan, Nigeria ($P = 2.13 \times 10^{-13}$ in Fisher's exact test). Because of potential applications of AI-PGx in precision population health, there is an unmet need to comprehensively identify AI-PGx in the human genome and to establish knowledge databases for differential populations and super-populations (super-populations are also called as continents in this paper).

Homozygosity disequilibrium (HD) is defined as a non-random pattern of a sizable run of homozygosity that its homozygosity intensity exceeds the equilibrium homozygosity intensity in the human genome[59]. Homozygosity disequilibrium analysis (HDA) has been used to study genetic ancestry, autozygosity, and natural selection in general population[60,61]; chromosomal aberrations responsible for complex disorders and cancers[59,60,62]; gene regulation of expression quantitative trait loci[63]; and pharmacogenomic and pharmacoepigenomic effects[64]. In contrast to single SNPs, the context of HD can serve as a multilocus AIM and AI-PGx to provide genetic information supplemental to single-locus AIMs and AI-PGx. In addition, HD that allows for a small proportion of heterozygotes due to genotype errors, mutations, and missing genotypes provides an analogous but more flexible homozygosity enrichment set than the conventional run of homozygosity that has been broadly applied to study genetic ancestry, demographic history, and disorders/traits in population genetics[65–74] and medical genetics[75–79].

The advent of parallel sequencing technologies[80–83] has drastically accelerated the discovery and application of single nucleotide variations (SNVs), including SNPs and rare variants (RVs). The 1000 Genomes Project[84] generated a rich whole-genome sequencing dataset consisting of more than 77 million SNVs in 2,504 individuals from 26 global populations representing five major continents—Africa (AFR), Americas (AMR), East-Asia (EAS), South-Asia (SAS), and Europe (EUR). Since then, several studies have used this dataset to investigate population genetics[34,84–87]. In this study, we examined the genomic structure, inferred demographic history, identified AIMs and ancestry-informative genes (AIGs), and established population prediction panels for global continents and populations by combining the valuable genomic resource of The 1000 Genomes Project with additional public pharmacogenetic databases. Based on these materials, we identified AI-PGx across the human genome, explored their features in pharmacogenomics, and established genetic ancestry pharmacogenomic databases.

## Results

We analyzed the whole-genome sequencing data of 2504 independent samples from 26 populations of five continental ancestry groups in The 1000 Genomes Project—Final Phase[84,88] (Fig. 1) and the PGx information from four public PGx resources: Drug Bank, PharmGKB, PharmaADME, and Biotransformation genes (Fig. 2a). The analysis flow of this study is provided (Fig. 3).

**Genomic structure in global continents and populations**. The ultrahigh-dimensional principal component analysis (PCA) (Supplementary Note 1) showed that, except for a few outliers, the African-ancestry, European-ancestry, East Asian-ancestry, and South Asian-ancestry groups were clearly separated (Fig. 4a), reflecting the differential genetic background of trans-ancestry groups. American-ancestry individuals were located in two middle clines (CLM and PUR in one cline and MXL and PEL in another cline), reflecting their disparate admixture proportions of the African-ancestry, European-ancestry, and Native-American-ancestry groups[89]. South Asian-ancestry individuals were located in the other cline, reflecting their ancient admixed genetic background from the European-ancestry, East Asian-ancestry, and Native Indian-ancestry groups[85].

The dendrogram of a hierarchical clustering analysis (HCA) further displays genetic distances among the studied ancestry continents and populations (Fig. 4b). The African-ancestry group had the largest genetic distance from other ancestry groups. This result, combined with molecular genetics and archeological evidence, confirmed that the African-ancestry group was the most ancient ancestry group. YRI and ESN in Nigeria in West Africa were joined first, connected to ACB (whose ancestors were traded from West Africa to the Caribbean in the early 16th Century), and then connected to GWD and MSL, which are located in other regions of West Africa. This lineage group was further connected to LWK in East Africa and finally combined with ASW. African diaspora ACB in the Caribbean and ASW in the southwestern US clustered in overlapping clines with long tails and some outliers, reflecting a large within-population genetic heterogeneity due to an admixture of African- and American-ancestry lineages. In the East Asian-ancestry group, CDX and KHV in Southeast Asia joined first, connected to CHB (North China) and CHS (South China) in East Asia, and finally combined with JPT in Northeast Asia. Although the South Asian-ancestry group was clearly separated from the East Asian-ancestry group, BEB from the eastern part of the Indian subcontinent were closer to the East Asian-ancestry lineage and closely linked to ITU and STU located in the south of the Indian subcontinent. These three populations separated from the European-ancestry-prone cline of PJL and GIH located in the northwest of the Indian subcontinent[85]. In the European-ancestry group, GBR and CEU in northern and western Europe were tightly connected, reflecting a shared genetic background. TSI and IBS in southern Europe were in nearby geographic regions and shared more genetic components but still separated in the principal component plot. FIN in northern Europe joined the European lineage at a late stage, showing a genetic discrepancy from other European populations. In the American-ancestry group, the first cline

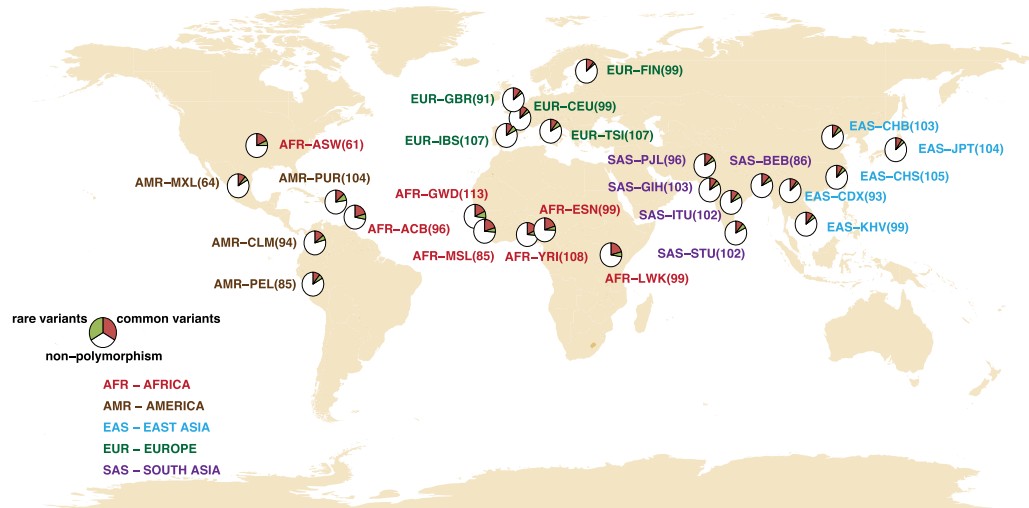

**Fig. 1 Twenty-six studied global populations from five continents.** The samples consisted of 661, 347, 504, 489, and 503 African-, American-, East Asian-, South Asian- and European-ancestry individuals, respectively. The 661 African-ancestry individuals consisted of 61 individuals of African ancestry in the Southwestern United States (ASW), 96 African Caribbean in Barbados (ACB), 113 Gambian in the Western Division, Mandinka (GWD), 99 Esan in Nigeria (ESN), 85 Mende in Sierra Leone (MSL), 108 Yoruba in Ibadan, Nigeria (YRI), and 99 Luhya in Webuye, Kenya (LWK). The 347 American-ancestry individuals consisted of 64 individuals of Mexican ancestry in Los Angeles, California, USA (MXL), 104 Puerto Ricans in Puerto Rico (PUR), 94 Colombians in Medellin, Colombia (CLM), and 85 Peruvians in Lima, Peru (PEL). The 504 East Asian-ancestry individuals consisted of 104 Japanese in Tokyo, Japan (JPT), 103 Han Chinese in Beijing, China (CHB), 105 Southern Han Chinese, China (CHS), 93 Chinese Dai in Xishuangbanna, China (CDX), and 99 Kinh in Ho Chi Minh City, Vietnam (KHV). The 489 South Asian-ancestry individuals consisted of 96 Punjabi in Lahore, Pakistan (PJL), 86 Bengali in Bangladesh (BEB), 103 Gujarati Indians in Houston, Texas, USA (GIH), 102 Indian Telugu in the UK (ITU), and 102 Sri Lankan Tamil in the UK (STU). The 503 European-ancestry individuals consisted of 99 Utah residents (CEPH) with northern and western European ancestry (CEU), 107 individuals from Iberian populations in Spain (IBS), 91 British in England and Scotland (GBR), 99 Finnish in Finland (FIN), and 107 Toscani in Italia (TSI).

consisting of populations PUR and CLM was closer to the lineage of the European-ancestry group, and the second cline consisting of populations MXL and PEL were closer to the lineage of the Native American-ancestry group[89]. These analyses revealed transcontinental genetic differentiation and cross-population genetic similarity and variability.

**Homozygosity disequilibrium in global continents and populations.** Genomic profiling of homozygosity intensity (Fig. 5) and population structure of HD in global continents and populations (Supplementary Fig. 1) are shown. According to the distributions of the number and width of genomic regions under HD in global continents (Fig. 6a) and populations (Fig. 6b), the African-ancestry group had the lowest number and shortest regions of HD. By contrast, the East Asian-ancestry group had the largest number and widest regions of HD. The American-ancestry group exhibited the largest spread in the number and length of regions of HD. All continents contained a certain proportion of individuals who carried extremely long regions under HD.

In general, the pattern of HD differed across the continental ancestry groups; 84.358% of regions of HD showed a significant difference in medians of homozygosity intensity among different continental ancestry groups. Moreover, compared with other continents, the American-ancestry group exhibited the largest within-continent variation in HD. Among the populations within the American-ancestry group, 0.740% ($N = 575,750$) of regions of HD exhibited significantly different medians of homozygosity intensity. In the non-American-ancestry groups, the proportions were much lower: 0.006% ($N = 4951$), 0.037% ($N = 16,960$), 0.037% ($N = 28,417$), and 0.006% ($N = 4731$) in AFR, EAS, EUR, and SAS, respectively. The HDA revealed differential genomic structures and demographic histories of the studied ancestry groups.

**Ancestry-informative markers.** Venn diagrams show that the distributions of AIMs based on different types of variation SNP, RV, SNP/RV, and Total SNV differed in various ancestry groups (Fig. 7a–d). The numbers and proportions of AIMs (i.e., $P_{AIM}$) are provided (Table 1). Across the human genome, 36.78% of SNVs were continental AIMs that could tell the between-continent differences in allele frequency. According to different types of SNVs, 99.52% of SNPs, 12.45% of RVs, and 99.94% of SNPs and/or RVs (i.e., the variant was an SNP in some continents but becomes an RV in other continents) were continental AIMs. In addition, as expected, the number of AIMs that could distinguish the within-continental differences in allele frequency were much fewer. Within the African-, American-, East Asian-, European-, and South Asian-ancestry groups, only 9.84%, 13.90%, 6.89%, 3.77%, and 0.70% of SNVs could distinguish the within-continent differences in allele frequency, respectively; in details, 24.51%, 52.60%, 23.24%, 11.93%, and 3.00% of SNPs were within-continental AIMs; 0% of RVs were within-continental AIMs; and 21.52%, 7.35%, 19.52%, 10.39%, and 4.10% of SNPs and/or RVs were within-continental AIMs. The Venn diagrams of the within-continental AIMs of each of the five study ancestry groups are displayed in Fig. 7 for whole-genome SNVs (Fig. 7a), SNPs (Fig. 7b), RVs (Fig. 7c), and SNPs and/or RVs (Fig. 7d). This demonstrated that 1186 SNVs, 2572 SNPs, 0 RVs, and 10 SNPs and/or RVs could serve as within-continental AIMs for all five studied continental ancestry groups. These results summarized the genetic ancestry informativeness contained in different types of genetic variation and provided a catalog of AIMs for global continents and populations.

**Ancestry-informative PGx and enrichment of AIMs in PGx.** Among 3259 autosomal PGx in this study, the top three major drug categories of PGx belonged to PD ($N = 2303$), PK ($N = 503$), and CYP endogenous substrates ($N = 334$) (Fig. 2b). The top three

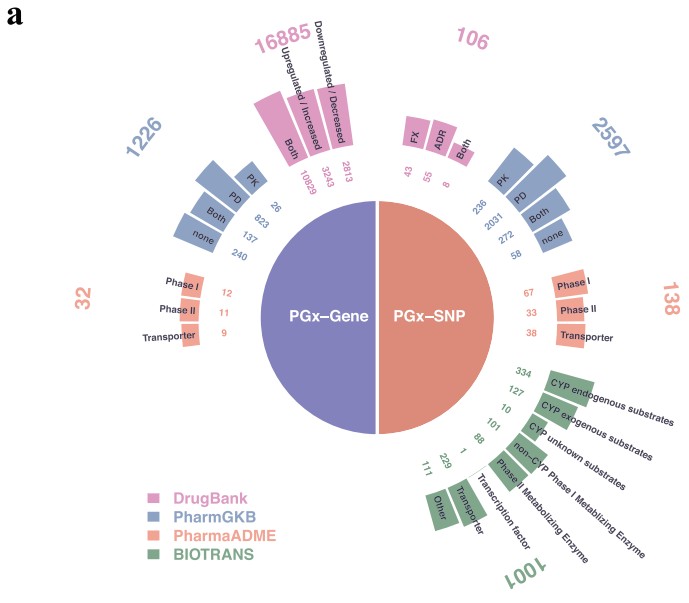

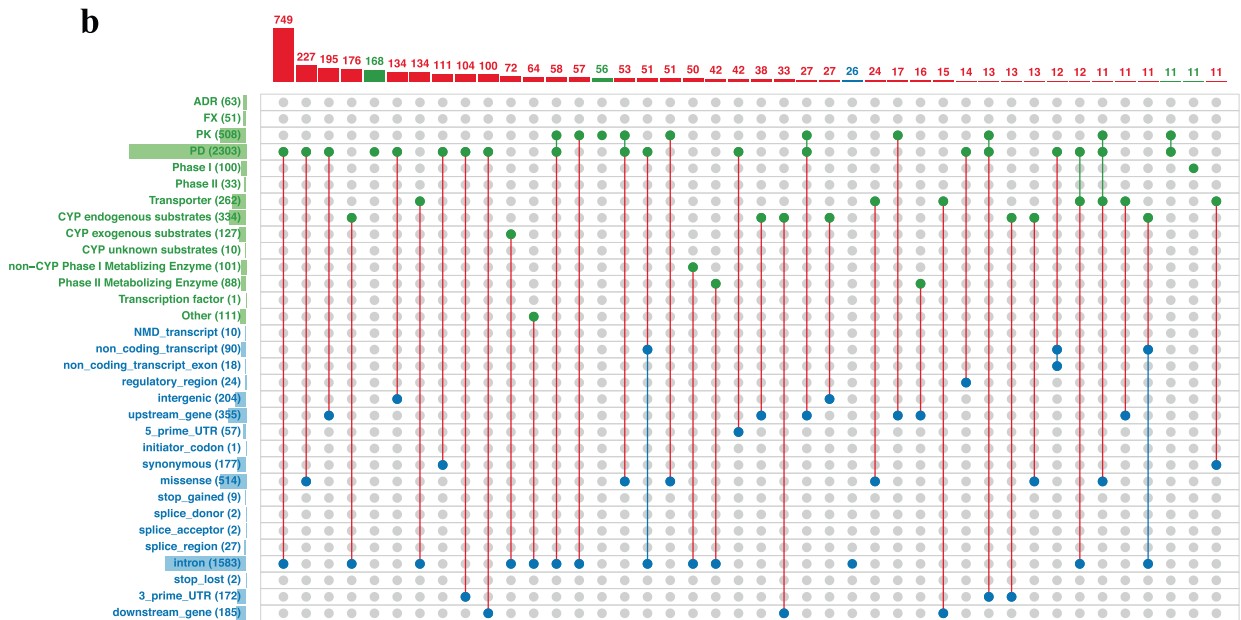

**Fig. 2 Distribution of PGx in four public resources. a** This study collected PGx from four PGx resources: Drug Bank, PharmGKB, PharmaADME, and Biotransformation. **b** Relationship between drug category and functional annotation of PGx. Red lines indicate a relationship between drug category and functional annotation. Green lines indicate a relationship between multiple drug categories. Blue lines indicate a relationship between multiple functional annotations. Histograms of PGx for drug category (green bar) and functional annotation (blue bar) are displayed on the left-hand side. Histograms of PGx for all relationships are displayed at the top; only the relationships with a frequency of >10 are shown.

major categories of functional annotation of PGx were intron ($N = 1583$), missense ($N = 514$), and gene upstream ($N = 355$) (Fig. 2b). For ancestry-informative PGx, the top three major drug categories and functional annotation categories of PGx remained the same. The results provided a complete catalog of ancestry-informative PGx characterized by different drug categories and functional annotation. Both protein-coding variants (e.g., a missense mutation) and non-protein-coding variants (e.g., an intronic SNV) represented important mechanisms for PGx.

Based on all autosomal SNVs, we calculated the proportion of AIMs (in notation, $P_{AIM}$) and the proportion of AIMs only based on PGx (in notation, $P_{AIM | PGx}$) in each of the six analysis groups (Table 2). The results showed that ($P_{AIM}$, $P_{AIM | PGx}$) = (27.97%,

98.04%), (8.96%, 26.97%), (12.40%, 39.89%), (6.45%, 16.66%), (3.63%, 9.30%), and (0.70%, 2.06%) in the whole-continental, African-, American-, East Asian-, European-, and South Asian-ancestry groups, respectively. One-sided Fisher's exact tests showed that $P_{AIM | PGx}$ was significantly higher than $P_{AIM}$ with a $P$ value of 0, $2.88 \times 10^{-195}$, 0, $7.93 \times 10^{-90}$, $5.25 \times 10^{-48}$, and $2.71 \times 10^{-14}$ in the whole-continental, African-, American-, East Asian-, European-, and South Asian-ancestry groups, respectively. The results revealed that AIMs are highly enriched in PGx loci, especially in ancestry groups with a high genetic differentiation.

**Ancestry informative genes**. Manhattan plots from a genome-wide homozygosity association study are displayed (Fig. 8). Among

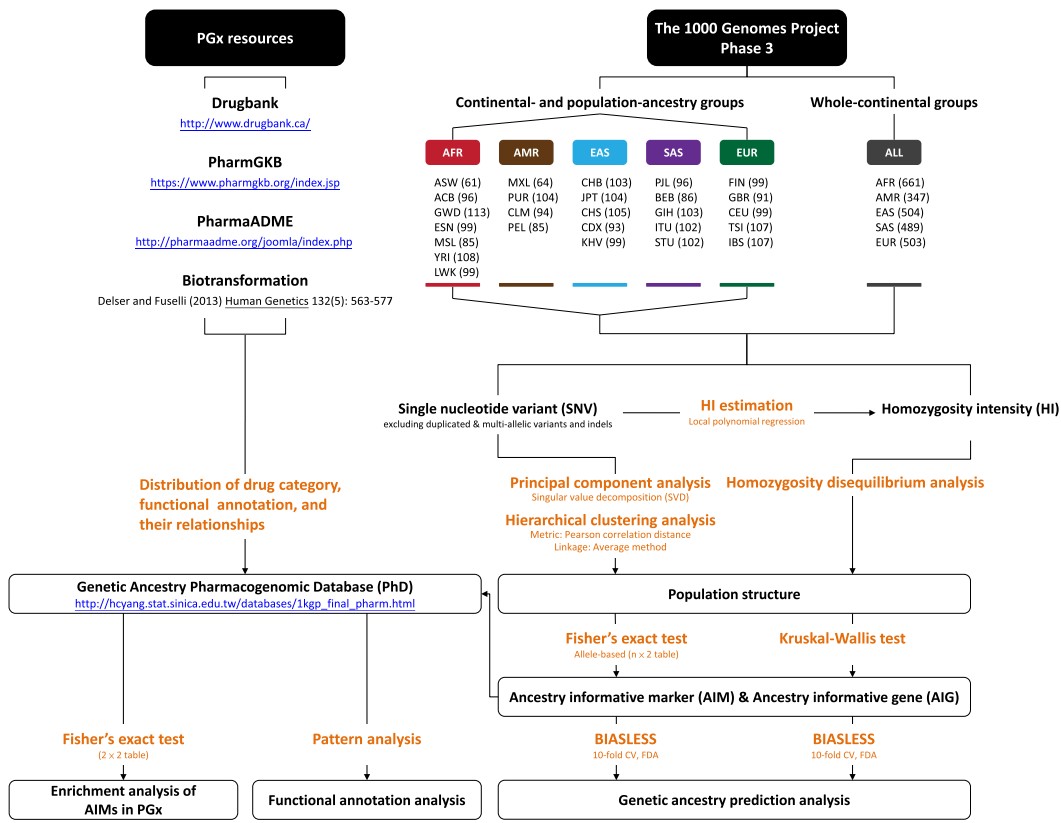

**Fig. 3 Analysis flow of this study.** Genomic structures of 26 populations in five continents were explored using principal component analysis, hierarchical clustering analysis, and homozygosity disequilibrium analysis. Within each of the six study ancestry groups (whole-continents, AFR, AMR, EAS, EUR, and SAS), AIMs were identified based on each of four genetic variation categories using two-sided Fisher exact tests. Ancestry-informative PGx were identified and an enrichment of AIMs in PGx loci was evaluated using one-sided Fisher exact tests. Finally, a genetic ancestry PGx database was constructed.

37,049 gene regions, we found 99.360% of AIGs ($N = 36,812$) under HD among continents. In specific continent, we found 2.904% ($N = 1076$), 34.562% ($N = 12,805$), 4.802% ($N = 1779$), 2.440% ($N = 904$), and 0.432% ($N = 160$) AIGs under HD in AFR, AMR, EAS, EUR, and SAS, respectively. The results suggested a pattern of differential autozygosity in global continents and populations and provided a catalog of AIGs under HD.

**Genetic ancestry prediction panels**. Based on 3002 autosomal SNVs, which were the within-continental AIMs for all of the five study ancestry groups (Fig. 7a–d) and three genes which were the within-continental AIGs under HD for all five study ancestry groups (Fig. 7e), a flexible discriminant analysis of 10-fold cross validation established the best discriminant model for classifying individuals to the five studied continental ancestry groups (Supplementary Fig. 2). The classification model obtained a training accuracy of 96.6% and a testing accuracy of 95.6% in correctly assigning individuals to their continental ancestry group (Supplementary Fig. 3). Genotype distributions of the 31 AIMs and 1 AIG (*LOC100132249*) are summarized in a staked-bar/box-whisker plot (Supplementary Fig. 4). The impact of individual predictive AIMs is shown in a marker impact plot (Supplementary Fig. 5)—The top three AIMs, consistently chosen in a 10-fold cross-validation analysis, were rs1578060 on *CCDC14*, rs5017562 on *CENPW*, and rs2682248 on *KIF26B*. All three genes were AIG genes. The result of sample clustering is displayed in a multi-dimensional scaling plot (Supplementary Fig. 6). The analysis established a prediction panel that can be used to determine the continent for an individual based on an inference of genetic ancestry.

**Pharmacogenomic database**. We established genetic ancestry pharmacogenomic database Genetic Ancestry PhD according to the study ancestry groups (AFR, AMR, EAS, EUR, SAS, and whole-continents) and four genetic variation categories (All SNVs, SNPs, RVs, and SNPs and/or RVs) at the website http://hcyang.stat.sinica.edu.tw/databases/genetic_ancestry_phd/. An instruction on how to use the Genetic Ancestry PhD is provided (Supplementary Note 2).

**Discussion**

Our population genomics analyses, including PCA, HCA, and HDA, revealed differential genetic and demographic backgrounds of the studied global continents and populations. For the African-ancestry group, PCA and HCA showed this group had the largest genetic distance from other continental ancestry groups and HDA showed this group had the lowest number and shortest regions of HD, which was also found in previous studies with smaller numbers of samples, ethnic populations, and genetic markers[60,74]. Combined with molecular genetics and archeological evidence, these results revealed that this group had a larger effective population size and was more ancient than other continental ancestry groups. In contrast to the indigenous residents in Africa, the African-ancestry diaspora populations exhibited a much more diverse genetic admixture. For the American-ancestry group, PCA and HCA showed two clearly separate clines (PUR–CLM and MXL–PEL) and HDA showed that this group exhibited the largest spread in the number and length of regions of HD among the studied ancestry groups. This revealed a large genetic differentiation and complex admixture in the Americas in this data set. For the South Asian-ancestry group, PCA showed that individuals from each population had been

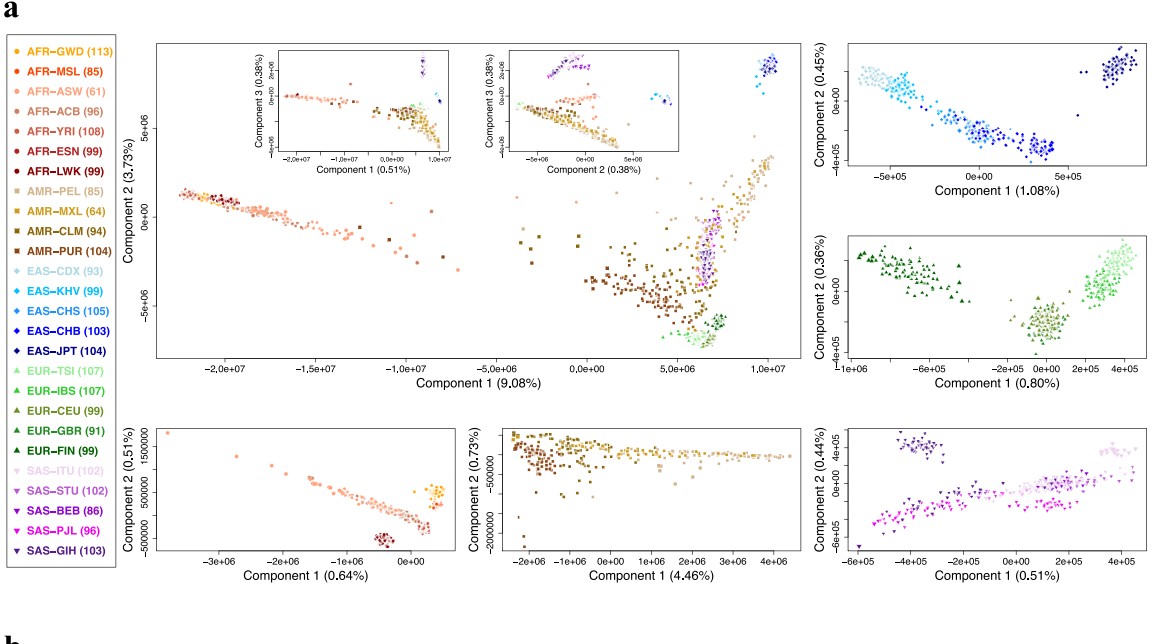

**Fig. 4 Global distribution of the whole-genome allele frequency. a** Principal component plots of whole-genome allele frequency. **b** Hierarchical clustering dendrogram of whole-genome allele frequency. Sample size n = 2504 individuals.

divided into two subgroups. This special substructure was also observed in previous research and was explained by a social hierarchy of individuals in the Indian subcontinent[85]. More investigations are warranted to clarify the intrinsic reasons for this special pattern. HDA found a number of outlier individuals who carried many more or larger regions of HD in this group. For the East Asian-ancestry group, PCA showed the Northeast Asian population JPT was separated from others, starting from East Asian populations (CHB and CHS) to Southeast Asian populations (CDX and KHV) along the latitudinal cline. HDA showed that they carried the largest number and widest regions of HD among the studied ancestry groups. This may be partially

explained by a recent bottleneck in East Asia[90]. For the European-ancestry group, PCA and HCA exhibited a latitudinal cline of Northern Europe (FIN) to Western Europe (GBR and CEU) to Southern Europe (IBS and TSI). HDA showed the distributions of the number and length of regions of HD were in between those of the East Asian-ancestry and South Asian-ancestry groups. In each continental ancestry group, we found a number of individuals who exhibited an extremely long genomic segment or multiple genomic segments under HD. For example, a 410-Mb region under HD was observed in individual HG04070 (ITU) (Supplementary Fig. 7a). Forty-seven regions under HD with a length of >2 Mb were found in individual HG02684 (CEU) (Supplementary

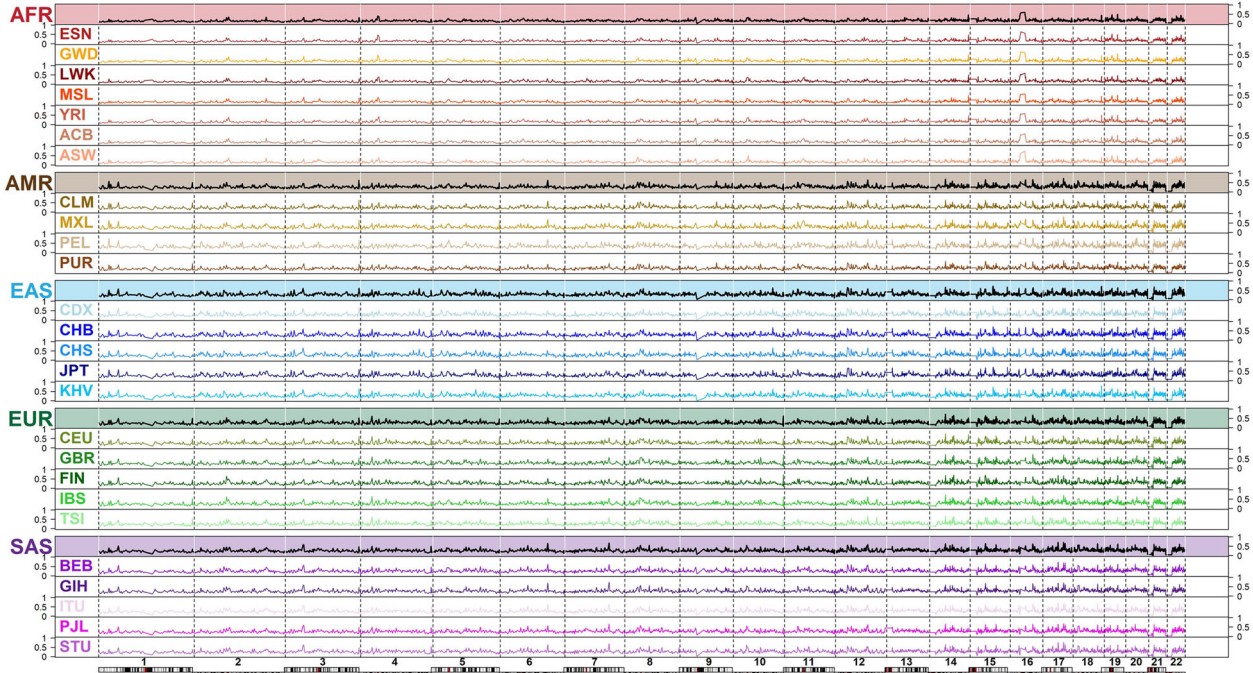

**Fig. 5 Genomic profiles of homozygosity intensity in global continents and populations.** Sample size $n = 2504$ individuals.

Fig. 7b). These patterns revealed that autozygosity was not a rare event in global continents and populations.

In this study, we found that East Asians carried the largest number and widest regions of HD in The 1000 Genomes Project. This finding is not in conflict with the previous finding[71,74] that Native Americans had the most recent and severe bottlenecks, which made the overall lengths and numbers of runs of homozygosity much higher in Americans than other populations. The American-ancestry populations in The 1000 Genomes Project studied in this paper differ from the populations in the Human Genome Diversity Panel[91,92] data. In the Human Genome Diversity Panel, the American-ancestry populations (Maya, Pima, Colombian, Karitiana, and Surul) are Native Americans. These Native Americans population had undergone recent and severe bottlenecks and exhibited much higher lengths of runs of homozygosity than other populations (Refer to Fig. 3 in Pemberton et al.[71]). In The 1000 Genomes Project, the American-ancestry populations (MXL, PUR, CLM, and PEL) are admixed Americans. This ancestry admixture reflects in the large variability of the lengths and numbers of HD (Refer to Fig. 6 in this paper). MXL was included in both of The 1000 Genomes Project and the International Haplotype Map Project III[93]. This admixed Americans population did not show higher lengths of runs of homozygosity compared to East Asians (Refer to Fig. 3 in Pemberton et al.[71]), and this result is consistent to our finding. The CLM participants in The 1000 Genomes Project were Colombians with admixed ancestry recruited from the second-largest city in Colombia and they differed from the Colombian participants with the Native Americans ancestry in the Human Genome Diversity Panel. As expected, the Colombian participants in the Human Genome Diversity Panel exhibited much higher lengths of runs of homozygosity compared to the CLM in The 1000 Genomes Project (Refer to Fig. 3 in Pemberton et al.[71]).

This study generated catalogs of AIMs and AIGs that provide genetic ancestry information for global continents and populations and established a prediction panel for global continents. We found more than 28 million among-continental AIMs and 0.17–3.83 million within-continental AIMs with the African-ancestry group having the most AIMs and the South Asian-ancestry group having the least AIMs. This reflected the genetic diversity in these continents. In addition, we found 36,812 AIGs under HD among continents and 160–12,805 AIGs under HD within continents, with the American-ancestry group having the most AIGs under HD within continents in reflection of the spread of genetic admixture in this continent. Furthermore, we established a marker panel that can be used to predict continental ancestry groups with a high accuracy according to the information of genetic ancestry from 31 AIMs and 1 AIG. Compared with previous prediction panels, we combine both of AIMs and AIGs under HD from whole-genome sequencing data to establish prediction panels for global continents.

Compared to the three available commercial panels[94], our postulated genetic ancestry prediction panel is either more accurate or parsimonious. Our panel that contains 31 AIMs and 1 AIG obtained a training accuracy of 96.6% and a testing accuracy of 95.6% in correctly assigning individuals to their continental ancestry group. The QIAGEN140-SNP Identification Multiplex panel[94] required a larger number of SNPs (140 SNPs) and had a lower training accuracy of 95.87% and a testing accuracy of 92.03% than ours. The Ion AmpliSeq HID Phenotyping Panel[95] used the 24 SNP HIrisplex System and had a notably reduced training accuracy of 81.1% and testing accuracy of 85.2% than ours. The 165 SNP Precision ID Ancestry Panel constituted by 123 SNPs from the group of M. Seldin[24] and 55 SNPs from the group of K. Kidd[96] with 13 overlapping SNPs, had a higher training accuracy of 99.29% and testing accuracy of 100%, but this panel required more SNPs than ours.

We observed a substantially high enrichment of AIMs in PGx. This study finds this informative phenomenon through a formal analysis of whole-genome sequencing data and pharmacogenomic data, and it illustrates that differential drug responses, ADRs, and drug's effective doses among ancestry groups are related to genetic ancestry. This explains the necessity to differentiate medical treatments according to not only population ancestry but also the individual's genetic ancestry and genotype information. Genetic ancestry plays a critical role in precision

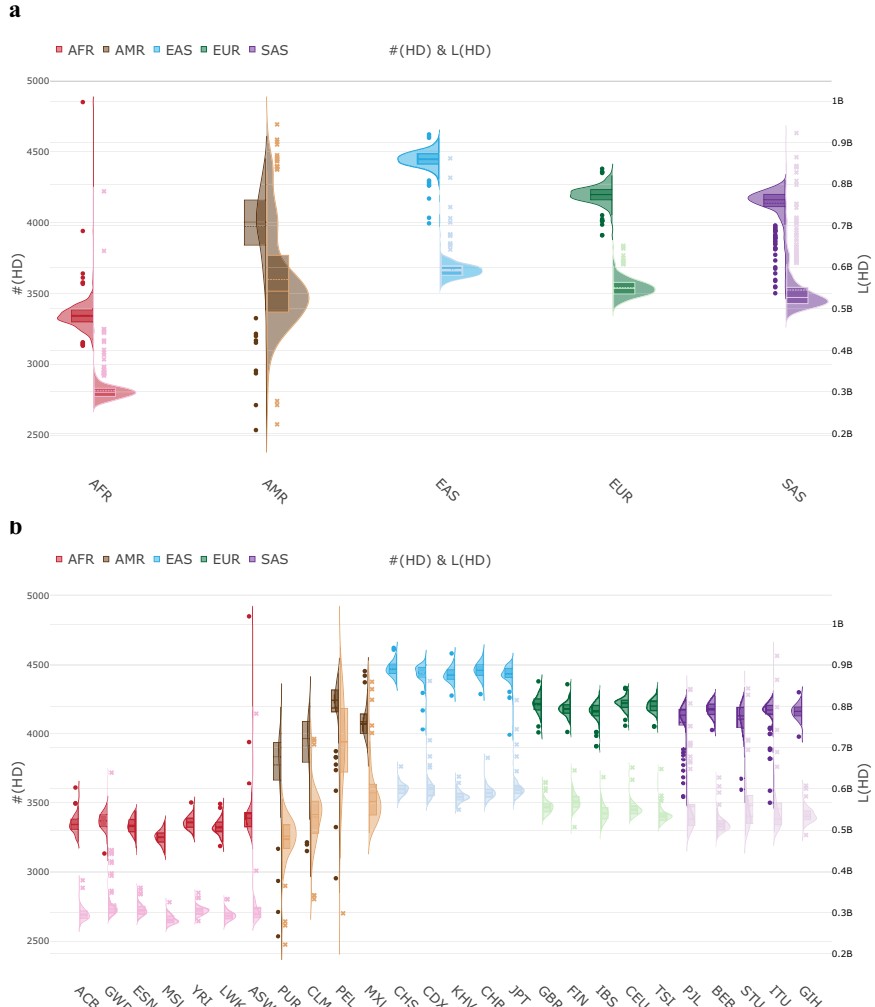

**Fig. 6 Genome-wide distributions of the number and length of regions of HD. a** Distributions of HD in the five continental ancestry groups.
**b** Distributions of HD in all populations in the five continental ancestry groups. For each continental ancestry group or population, the number of regions of HD, i.e., #(HD), is displayed on the left-hand side of a violin plot with respect to the left y-axis. The length of regions of HD, i.e., L(HD), is displayed on the right-hand side of a violin plot with respect to the right y-axis. Sample size $n = 2504$ individuals.

population health and the development of gene therapies and cancer testing and should include more population groups with different genetic backgrounds to benefit more people around the world.

Our pharmacogenomic database revealed that an AI-PGx with multiple drug targets and differential allele frequencies among global continents was often located in protein-coding regions (e.g., missense and synonymous variants) and related to important biological function(s). For example, rs1801133 (i.e., C677T and Ala222Val) is a $C > T$ missense SNP on the methylenetetrahydrofolate reductase (MTHFR) gene at 1p36.22. MTHFR is known to be associated with catalysis of the conversion from 5,10-methylenetetrahydrofolate to 5-methyltetrahydrofolate and folate metabolism. Allele T leads to a valine substitution at amino acid 222 and encodes a thermolabile MTHFR enzyme with reduced activity. Methylenetetrahydrofolate reductase deficiency decreases concentrations of folate and red blood cells and alters susceptibility to multiple diseases, such as hyperhomocysteinemia, occlusive vascular disease, neural tube defects, colon cancer, and acute leukemia. Drugs targeted by this AI-PGx include methotrexate, vitamin B-complex, and so on. Supplementation with folate for individuals who carry homozygote TT can improve methylenetetrahydrofolate reductase deficiency.

This study showed that rs1801133 had a significant difference in allele frequency among global continents (adjusted $P = 3.568 \times 10^{-4}$ in Fisher's exact test) and the maximum allele frequency difference among the five studied continents was $\Delta = 0.384$; the allele frequencies of allele T were 0.090, 0.119, 0.296, 0.365, and 0.474 in AFR, SAS, EAS, EUR, and AMR, respectively (Supplementary Fig. 8). The frequency reflects the prevalence of methylenetetrahydrofolate reductase deficiency in different ancestry groups. In addition to rs1801133, other examples of AI-PGx with multiple drug targets included but were not limited to missense rs1801131 on MTHFR (adjusted $P = 3.568 \times 10^{-4}$ in Fisher's exact test, $\Delta = 0.266$), synonymous rs6305 on HTR2A (adjusted $P = 4.24 \times 10^{-12}$ in Fisher's exact test, $\Delta = 0.0259$), missense rs6025 on F5 (adjusted $P = 3.14 \times 10^{-6}$ in Fisher's exact test, $\Delta = 0.0119$), and missense rs149157808 (merged to rs28371733) on CYP2D6 (adjusted $P = 5.46 \times 10^{-6}$ in Fisher's exact test, $\Delta = 0.0121$). Drug targets and allele frequency distributions of the aforementioned AI-PGx are provided in our pharmacogenomic database.

In addition to a significant difference in allele frequency among global continents, AI-PGx with target multiple drugs may also exhibit significantly differential allele frequencies within specific continent(s). For example, missense rs1801133 also had

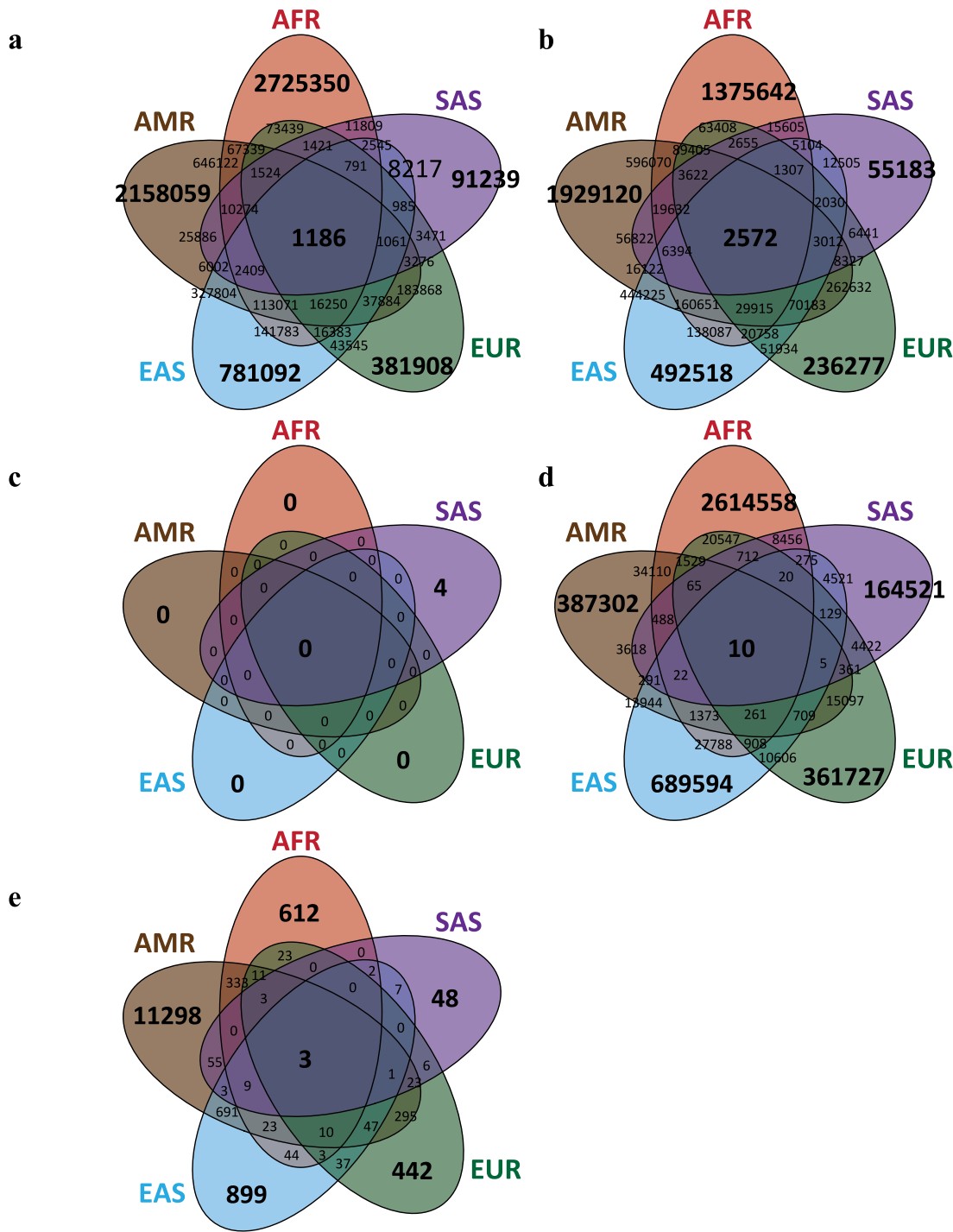

**Fig. 7 Distribution of ancestry informative markers (AIMs) and ancestry informative genes (AIGs). a** SNV-based AIMs. **b** SNP-based AIMs. **c** RV-based AIMs. **d** SNP/RV-based AIMs. **e** AIGs under homozygosity disequilibrium (HD).

significantly different allele frequencies within the East Asian-ancestry group (adjusted $P = 4.157 \times 10^{-3}$ in Fisher's exact test, $\Delta = 0.332$) and significantly different allele frequencies within the European-ancestry group (adjusted $P = 7.479 \times 10^{-3}$ in Fisher's exact test, $\Delta = 0.195$). In the East Asian-ancestry group, the allele frequencies of $T$ were 0.134, 0.192, 0.286, 0.380, and 0.466 in CDX, KHV, CHS, JPT, and CHB, respectively; the maximum difference was $\Delta = 0.332$. In the European-ancestry group, the allele frequencies of $T$ were 0.727, 0.702, 0.676, 0.556, and 0.533 in

FIN, CEU, GBR, IBS, and TSI, respectively; the maximum difference was $\Delta = 0.195$. These examples illustrate again that, if the information of genetic ancestry and individual genotype is available, they should be considered in the use of medication. Other examples include: (1) in AFR—rs2231142 on *ABCG2* (adjusted $P = 8.592 \times 10^{-4}$ in Fisher's exact test, $\Delta = 0.0738$), rs1799971 on *OPRM1* (adjusted $P = 1.949 \times 10^{-3}$ in Fisher's exact test, $\Delta = 0.0492$), and rs4149056 on *SLCO1B1* (adjusted $P = 3.748 \times 10^{-3}$ in Fisher's exact test, $\Delta = 0.0656$); (2) in AMR—

rs738409 on *PNPLA3* (adjusted $P = 1.94 \times 10^{-12}$ in Fisher's exact test, $\Delta = 0.163$), and rs11854484 on *SLC28A2* (adjusted $P = 6.95 \times 10^{-11}$ in Fisher's exact test, $\Delta = 0.306$); (3) in EAS—rs4961 on *ADD1* (adjusted $P = 4.157 \times 10^{-3}$ in Fisher's exact test, $\Delta = 0.144$); (4) in EUR—rs1137101 on *LEPR* (adjusted $P = 7.479 \times 10^{-3}$ in Fisher's exact test, $\Delta = 0.152$), rs1801394 on *MTRR* (adjusted $P = 7.479 \times 10^{-3}$ in Fisher's exact test, $\Delta = 0.111$), and rs4149117 and rs7311358 on *SLCO1B3* (adjusted $P = 7.479 \times 10^{-3}$ in Fisher's exact test, $\Delta = 0.163$); (5) in SAS—rs6280 on *DRD3* (adjusted $P = 2.64 \times 10^{-2}$ in Fisher's exact test, $\Delta = 0.219$) and rs2229774 on *RARG* (adjusted $P = 2.64 \times 10^{-2}$ in Fisher's exact test, $\Delta = 0.185$).

Top AIMs with the maximum difference in allele frequencies across the studied continental ancestry groups and within a specific continental ancestry group are also provided in our pharmacogenomic database (Supplementary Note 3). Previous studies on AIMs have mainly focused on single-locus AIMs. We further explored the topic by examining AIGs under HD and developing a catalog of AIGs under HD for global continents and populations. It is not surprising that the whole-continental genome-wide homozygosity association analysis identified an enormous number of AIGs under HD across the human genome ($N = 36,812$). The American-ancestry group exhibited a substantial number of AIGs under HD ($N = 12,805$), which reflects the heterogeneous genetic background of this continental ancestry group in this data set.

We found that the distributions of HD in PGx regions coincided with the patterns of HD in the whole genome, but the discrepancies of homozygosity intensities shrunk among the studied continental and population ancestry groups. In addition, our HDA identified AIGs with reported evidence of natural selection such as the *CYP3A* family[97,98], *LCT*[99,100], and *MCM6*[100,101]. For example, the *CYP3A* family contains four main genes: *CYP3A4*, *CYP3A43*, *CYP3A5*, and *CYP3A7*. The encoded isozyme plays a critical role in the metabolism of therapeutics, endogenous metabolites (e.g., hormones and antibiotics), exogenous chemicals (e.g., environmental contaminants and food additives), and salt homeostasis[97]. Our database showed that all *CYP3A* family members exhibited significantly different levels of homozygosity intensity across the five studied continental ancestry groups; *CYP3A4* (adjusted $P = 4.471 \times 10^{-48}$ in the Kruskal–Wallis test; median homozygosity intensity ranges from 0.757 to 0.909), *CYP3A43* (adjusted $P = 1.828 \times 10^{-65}$ in the Kruskal–Wallis test; median homozygosity intensity ranges from 0.718 to 0.879), *CYP3A5* (adjusted $P = 4.971 \times 10^{-40}$ in the Kruskal–Wallis test; median homozygosity intensity ranges from 0.811 to 0.965), and *CYP3A7* (adjusted $P = 5.061 \times 10^{-49}$ in the Kruskal–Wallis test; median homozygosity intensity ranges from 0.793 to 0.975).

Some examples demonstrated a high selective pressure in continental ancestry groups and specific population ancestry groups. For example, *Lactase* (*LCT*) on 2q21.3 encodes an enzyme that aids in lactose digestion. A lactase deficiency results in a lactose intolerance that has an autosomal recessive mode of inheritance and is associated with homozygosity of the gene. The prevalence of lactose intolerance has a wide global spectrum and Caucasians have a lower prevalence than that of other populations[102,103]. *LCT* was also identified as an AI-PGx responsible for decitabine. Our results showed a significant difference in homozygosity intensities across the continental ancestry populations (adjusted $P = 9.057 \times 10^{-34}$ in the Kruskal–Wallis test) and a significant difference in homozygosity intensities in the European-ancestry group (range of homozygosity intensities $\Delta = 0.422$, adjusted $P = 2.707 \times 10^{-4}$ in the

### Table 1 Global distributions of AIMs.

| Ancestry groups | VarType | # of SNVs | # of AIMs based on allele-based 2-sided Fisher exact test (the proportion of AIMs, $P_{AIM}$) False discovery rate |
|---|---|---|---|
| ALL | MAF = 0 | 245,462 | - |
|  | SNP | 5,533,074 | 5,506,536 (99.520%) |
|  | RV | 58,354,524 | 7,264,275 (12.449%) |
|  | SNP/RV | 13,685,123 | 13,677,393 (99.944%) |
|  | Total SNV* | 77,572,721 | 28,531,199 (36.780%) |
| AFR | MAF = 0 | 38,873,037 | - |
|  | SNP | 10,326,460 | 2,530,827 (24.508%) |
|  | RV | 16,020,578 | 0 (0%) |
|  | SNP/RV | 12,598,108 | 2,711,122 (21.520%) |
|  | Total SNV* | 38,945,146 | 3,831,696 (9.839%) |
| AMR | MAF = 0 | 51,903,729 | - |
|  | SNP | 7,032,091 | 3,698,704 (52.597%) |
|  | RV | 12,632,724 | 0 (0%) |
|  | SNP/RV | 6,249,639 | 459,185 (7.347%) |
|  | Total SNV* | 25,914,454 | 3,602,015 (13.900%) |
| EAS | MAF = 0 | 56,021,288 | - |
|  | SNP | 6,271,954 | 1,457,317 (23.235%) |
|  | RV | 11,680,819 | 0 (0%) |
|  | SNP/RV | 3,844,122 | 750,456 (19.522%) |
|  | Total SNV* | 21,796,895 | 1,501,008 (6.886%) |
| EUR | MAF = 0 | 55,685,350 | - |
|  | SNP | 7,161,303 | 854,478 (11.932%) |
|  | RV | 10,956,920 | 0 (0%) |
|  | SNP/RV | 4,014,610 | 417,108 (10.390%) |
|  | Total SNV* | 22,132,833 | 834,331 (3.770%) |
| SAS | MAF = 0 | 53,228,676 | - |
|  | SNP | 7,239,212 | 217,333 (3.002%) |
|  | RV | 12,766,736 | 4 (0.000031%) |
|  | SNP/RV | 4,583,559 | 187,916 (4.100%) |
|  | Total SNV* | 24,589,507 | 172,096 (0.700%) |

*Total SNV indicates all polymorphic variants with a nonzero minor allele frequency (MAF), including common variant (SNP), rare variant (RV), and SNP/RV, where 162 variants with duplicated positions were excluded.

### Table 2 Global distributions of AIM and distributions of AIM in pharmacogenomic loci.

| Continent | PGx ∩ AIM | PGx$^c$ ∩ AIM | PGx ∩ AIM$^c$ | PGx$^c$ ∩ AIM$^c$ | $P_{AIM}$ | $P_{AIM\,\mid\,PGx}$ | $p$ |
|---|---|---|---|---|---|---|---|
| ALL | 3195 | 28,528,004 | 64 | 73,462,356 | 0.2797 | 0.9804 | 0 |
| AFR | 879 | 3,830,817 | 2380 | 38,934,065 | 0.0896 | 0.2697 | $2.88 \times 10^{-195}$ |
| AMR | 1300 | 3,600,715 | 1959 | 25,454,504 | 0.1240 | 0.3989 | 0 |
| EAS | 543 | 1,500,465 | 2716 | 21,779,390 | 0.0645 | 0.1666 | $7.93 \times 10^{-90}$ |
| EUR | 303 | 834,028 | 2956 | 22,118,305 | 0.0363 | 0.0930 | $5.25 \times 10^{-48}$ |
| SAS | 67 | 172,029 | 3192 | 24,585,458 | 0.0070 | 0.0206 | $2.71 \times 10^{-14}$ |

*PGx$^c$ indicates non-PGx loci; AIM$^c$ indicates non-AIM loci; symbol ∩ indicates an intersection. $P_{AIM}$ indicates the proportion of AIMs; $P_{AIM\,\mid\,PGx}$ indicates the proportion of ancestry-informative loci in PGx.

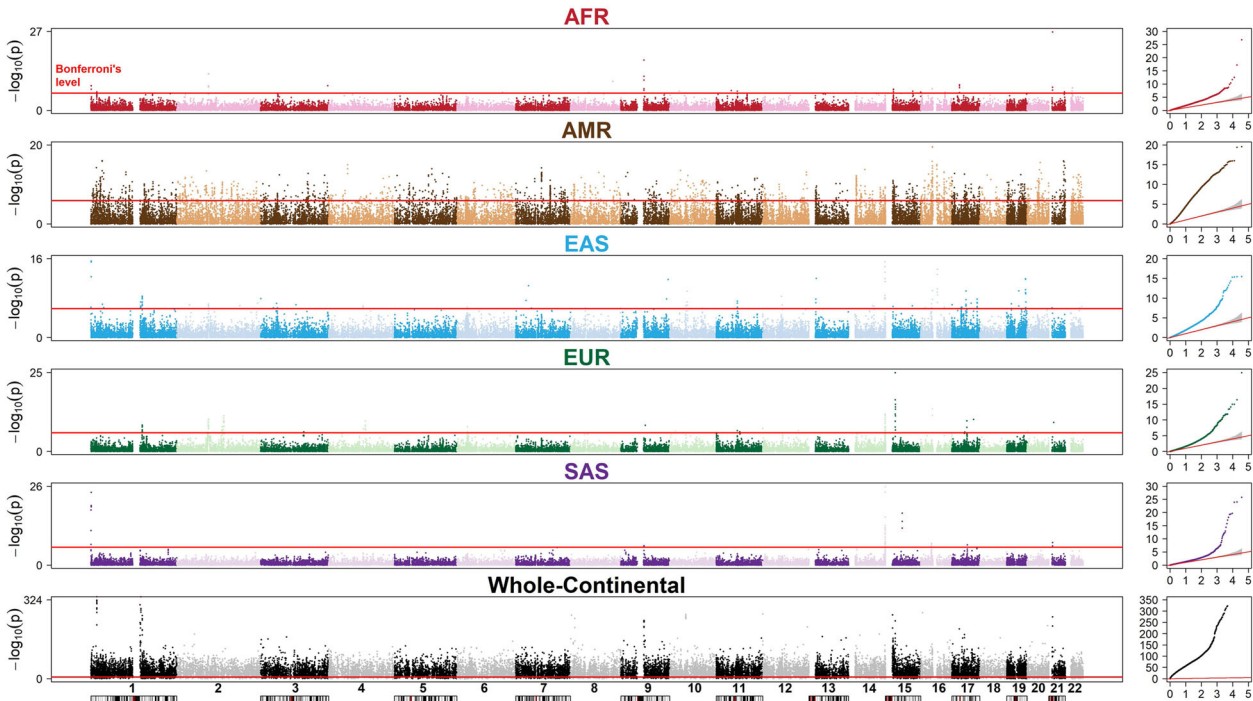

**Fig. 8 Genome-wide homozygosity association study.** The results of genome-wide homozygosity association tests for each of the five studied continental ancestry groups (AFR, AMR, EAS, EUR, and SAS) and for a comparison of the five continental ancestry groups in the whole-continental group (whole-continents) are shown in Manhattan plots. The vertical axis represents the P values (−log10 scale) of the homozygosity association tests based on a two-sided Kruskal–Wallis test. The horizontal axis represents the physical positions of the anchor SNPs of sliding windows by chromosome. A red reference line indicates a P value threshold of a Bonferroni multiple-testing correction. Quantile–quantile plots (Q–Q plot) are provided on the right-hand side. Sample size n = 2504 individuals.

Kruskal–Wallis test). CEU and GBR had strikingly high median homozygosity intensities (0.991 in CEU and 0.995 in GBR), which were also much higher than the values in other European-ancestry populations (0.573 in FIN, 0.634 in IBS, and 0.578 in TSI) and in other continental ancestry groups (0.753 in AFR, 0.637 in AMR, 0.641 in EAS, and 0.630 in SAS). The patterns of prevalence of lactose intolerance and homozygosity intensity therefore coincide.

The *Minichromosome maintenance complex component 6* (*MCM6*) gene on 2q21.3 is another key gene associated with lactose intolerance in early adulthood[104]. This AI-PGx is responsible for the PK/PD of multiple medications, including tropine, estradiol, and so on. Our results showed that similar to *LCT*, *MCM6* also exhibited strikingly high median homozygosity intensities that were considerably higher than the values in other European- and non-European-ancestry groups with 0.993 in CEU and 0.994 in GBR. Different lengths of homozygosity regions may reflect different population histories. A short segment of homozygosity in tens of KB may be formed by a pair of ancient haplotypes that contribute to local LD patterns[71,74]; however, it is not always the case. CEU and GBR did not exhibit a striking difference in LD patterns compared to other European- and non-European-ancestry groups in *LCT* and *MCM6*. A previous study demonstrated that *LCT* and *MCM6* acted as a selective pressure in the European-ancestry group[100]. This study further showed the detailed patterns of differential signatures of selective pressure in ancestry populations within this continent.

Both protein-coding and non-protein-coding regions are necessary for a complete population pharmacogenomics study. According to the four popular pharmacogenetic resources used in this study, the largest and second largest functional groups of PGx were intron (48.052% of all PGx; N = 1566) and missense (15.496% of all PGx; N = 505). However, the majority of previous population

pharmacogenetic studies focused only on PGx in protein-coding regions[53,105] or differentially expressed regions[106]. The studies provided biological interpretations to protein-coding PGx but failed to provide a whole picture of the PK/PD mechanisms. The current study provides the unbiased investigation of PGx by analyzing a whole-genome sequencing data set of global populations. In addition to protein-coding PGx, the identification of AI-PGx in non-protein-coding regions provides information supplemental to previous studies.

The current study focused on DNA sequencing data containing rich information in genetic ancestry, population genomics, and pharmacogenomics. Future research can integrate additional –omics data, including but not limited to transcriptomics, epigenomics, and proteomics, to further decipher the relationship between genetic ancestry and drug PK/PD and move forward toward a comprehensive understanding of precision population health.

## Methods

**Samples**. In this study, we analyzed the whole-genome sequencing data of 2504 independent samples from 26 populations of five continental ancestry groups as presented by The 1000 Genomes Project—Final Phase[84,88] (Fig. 1). The analysis considered the six analysis groups: (1) African-ancestry group containing populations ASW, ACB, GWD, ESN, MSL, YRI, and LWK; (2) American-ancestry group containing populations MXL, PUR, CLM, and PEL; (3) East Asian-ancestry group containing populations JPT, CHB, CHS, CDX, and KHV; (4) South Asian-ancestry group contained populations PJL, BEB, GIH, ITU, and STU; (5) European-ancestry group containing populations CEU, IBS, GBR, FIN, and TSI; and (6) whole-continental group containing African-, American-, East Asian-, South Asian-, and European-ancestry groups.

**Quality control and filtering**. All 2504 samples were sequenced across the whole genome using either Illumina HiSeq 2000 or Illumina HiSeq 2500 by The 1000 Genomes Project[88]. The sequencing experiments provided genotype data for 81,271,745 genetic variants on 22 pairs of human autosomes. The sequencing data

are publicly available on The 1000 Genomes Project website (http://www.1000genomes.org/data). The genetic variants were annotated based on the Genome Reference Consortium Human genome build 37 (GRCh37). After removing duplicate variants, multi-allelic variants, insertions, and deletions, 77,818,183 SNVs remained. We further removed SNVs with a minor allele frequency of 0; in total, 245,462, 38,873,037, 51,903,729, 56,021,288, 53,228,676, and 55,685,350 SNVs were removed in the whole-continental, African-ancestry, American-ancestry, East Asian-ancestry, South Asian-ancestry, and European-ancestry groups, respectively. In the subsequent analysis, SNVs were classified into four genetic variation categories: (1) All SNVs for SNVs with a nonzero minor allele frequency (covering all three other categories of SNVs together); (2) SNPs for SNVs with a minor allele frequency of >0.01; (3) RVs for SNVs with a nonzero minor allele frequency of ≤0.01; and (4) SNPs and/or RVs for SNVs with a nonzero minor allele frequency of ≤0.01 in some continental or population ancestry groups and of >0.01 in others.

**Public PGx resources and functional annotation**. This study used the PGx information from four public PGx resources (Fig. 2a): (1) Drug Bank (http://www.drugbank.ca/), which consisted of 106 PGx SNPs categorized as ADR only, FX only, or FX & ADR and 16,885 PGx genes categorized as downregulated, upregulated, or both (update: 02/04/2018); (2) PharmGKB (https://www.pharmgkb.org/), which consisted of 2597 PGx SNPs and 1226 PGx genes categorized as PK only, PD only, PK & PD, or non-PK & non-PD (update: 05/04/2018); (3) PharmaADME (http://www.pharmaadme.org/joomla/), which consisted of 138 PGx core SNPs and 32 PGx core genes categorized as Phase I, Phase II, or Transporter (update: 05/04/2018); and (4) Biotransformation genes, which consisted of 1001 PGx SNPs grouped into eight subcategories[45,49]. The combination of all PGx records in the four public PGx resources consisted of 3,259 and 16,221 distinct autosomal PGx loci and genes, respectively. Functional annotation of PGx from The 1000 Genomes Project was as follows: NMD transcript, noncoding transcript, noncoding transcript exon, regulatory region, intergenic, upstream gene, 5′ UTR, initiator codon, synonymous, missense, stop gained, splice donor, splice acceptor, splice region, intron, stop lost, 3′ UTR, and downstream gene. The frequencies of the relationships between drug category and functional annotation of PGx are provided (Fig. 2b). Here, the drug category was defined according to the relation of PGx and drug biotransformation and PK/PD.

**Statistics and reproducibility**

*Principal component analysis and hierarchical cluster analysis*. A global population genomic structure of 2504 individuals from 26 populations was explored using an ultrahigh-dimensional PCA. An efficient algorithm of the ultrahigh-dimensional PCA for a whole-genome sequencing dataset of SNV genotypes was developed[107] (Supplementary Note 1). All individuals in the whole-continental group or the individuals from each continental ancestry group were projected onto the subspace of the first two principal components based on a singular value decomposition of a variance–covariance matrix of whole-genome individual-level allele frequency data. An HCA using an average linkage was performed, and population structure was displayed using a hierarchical clustering dendrogram.

*Homozygosity disequilibrium analysis*. HD, originally coined by Yang et al[59], is defined by a non-random pattern of sizable run of homozygosity where its homozygosity intensity exceeds the value under equilibrium in the human genome. Homozygosity intensity can be estimated based on SNV genotypes and the estimate ranges between 0 and 1. A higher value of homozygosity intensity indicates a higher homozygosity in a genomic region. The procedures of our HDA are described as follows. First, the genome-wide profile of homozygosity intensity for every individual was calculated based on genotype data under a double-weight local polynomial model by using LOHAS version 2.3[60]. The double weights were composed of a cubic kernel weight for considering a local smoothing property and a locus weight with a threshold of minor allele frequency of 0.05 for adjusting for the low informativeness of RVs. The procedure was applied to each gene region to estimate the homozygosity intensity of a gene for each individual. Second, to draw genomic profiling of average homozygosity intensity over individuals in each gene, the homozygosity intensities for each population or for all populations within each of the six analysis groups (refer to the Samples subsection) were summarized by taking an average (median and mean) homozygosity intensity of all individuals in the population or analysis group, respectively. Third, for all gene regions and the gene regions that contain PGx, the genomic distributions of the number and length of the regions under HD (i.e., an average homozygosity intensity in a gene region is of >0.9) in each population and continental ancestry group were summarized in violin plots.

*Identification of ancestry informative markers*. This study identified AIMs in each of the six analysis groups on the basis of the four genetic variation categories, resulting in 24 analyses in total. In each of the 24 analyses, an average individual-level allele frequency of every SNV in each of the six analysis groups was calculated. Two-sided Fisher's exact test[108] was performed to examine whether SNVs had a significant difference in allele frequency among the studied populations in a continent and among the five-continent ancestry groups.

A false discovery rate[109] for multiple testing was performed to calculate the adjusted $P$ values. An SNV with an adjusted $P$ value of <0.05 was assigned as an AIM. The number of AIMs ($N_{AIM}$) and the proportion of AIMs ($P_{AIM}$) were calculated.

*Enrichment analysis of AIMs in PGx*. We examined enrichment of AIMs in PGx. Three statistics were calculated: (1) the number of PGx ($N_{PGx}$) based on the four studied PGx resources; (2) the number of ancestry-informative PGx ($N_{AI-PGx}$); and (3) the proportion of ancestry-informative loci in PGx ($P_{AIM \mid PGx} = N_{AI-PGx} / N_{PGx}$). The excess of $P_{AIM \mid PGx}$ compared with $P_{AIM}$ was examined using one-sided Fisher's exact test. Odds ratio and $P$ value were also calculated. If $P_{AIM} < P_{AIM \mid PGx}$, it indicated that AIMs were enriched in PGx. All of the aforementioned analyses were performed using R packages.

*Identification of ancestry informative genes under homozygosity disequilibrium*. Genome-wide profile of homozygosity intensity for every individual was obtained from the HDA. The two-sided Kruskal–Wallis test[110] was employed to examine whether a median homozygosity intensity in a gene was significantly different within each of the six analysis groups. Adjusted $P$ values were calculated by applying a false discovery rate[109] for multiple testing, and then Manhattan plots were drawn. A gene with an adjusted $P$ value of <0.05 was assigned as an AIG.

*Genetic ancestry prediction analysis*. Genetic ancestry prediction panel was developed by using BIASLESS version 1.0[30]. The procedures used are described as follows: First, the analysis was implemented using a 10-fold cross-validation procedure. All 2504 individuals from the five studied continents were randomly partitioned into ten subsets. In all subsets, the proportions of individuals belonging to the five studied continents were the same as the proportions in all 2504. Second, based on the individuals in the first nine subsets (i.e., the first training dataset), a flexible discriminant analysis was applied to build a classification model with the highest training accuracy by sequentially selecting the AIMs or AIGs with the maximum increment of training accuracy. The AIM or AIG with the minimum ratio of the within-continent and between-continent sum of squares for genotypic values was selected if more than one AIM or AIG had the same training accuracy. The procedure continued until the training accuracy reached 1.0 or the increment of training accuracy was less than 0.001. Third, the model with the highest training accuracy was then used to classify individuals into continents in the remaining 10th subset (i.e., the first testing dataset) and calculate a testing accuracy. Fourth, the previous steps were repeated until each of the ten subsets of data had been analyzed as a testing dataset, resulting in ten classification candidate models. Finally, among the ten classification models, the model with the highest testing accuracy was selected as the final classification model and the AIMs and AIGs formed a continental prediction panel.

**Reporting summary**. Further information on research design is available in the Nature Research Reporting Summary linked to this article.

## Data availability
The sequencing data are publicly available on The 1000 Genomes Project website (http://www.1000genomes.org/data). The PGx information in this study can be accessed from public PGx resources: Drug Bank (http://www.drugbank.ca/); PharmGKB (https://www.pharmgkb.org/); PharmaADME (http://www.pharmaadme.org/joomla/); Biotransformation[45,49]. All analyzed data is available from the authors upon request. This study analyzed the public-domain data and was approved by Institute Review Board on Biomedical Science Research, Academia Sinica (approval number: AS-IRB01-17025).

## Code availability
Code for our developed ultrahigh-dimensional PCA plot generator is deposited in Zenodo (https://doi.org/10.5281/zenodo.4301096).

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

## Acknowledgements

This work was supported by a research grant from the Ministry of Science and Technology of Taiwan (MOST 106-2314-B-001-003-MY3). We thank Drug Bank, PharmGKB, and PharmaADME for generating the drug information. We thank The 1000 Genomes Project for making their next-generation sequencing data available online.

## Author contributions

H.C.Y. conceived the study and prepared the manuscript. C.W.C. and Y.T.L. developed the codes and analyzed the data with H.C.Y. C.W.C. created the genetic ancestry pharmacogenomic database Genetic Ancestry Ph.D. S.K.C. contributed to the discussion about pharmacogenetic information. All authors read and approved the final manuscript.

## Competing interests

The authors declare no competing interests.
