## [Peer Review File · Communications Biology]

Reviewers' comments:

Reviewer #1 (Remarks to the Author):

This is an interesting manuscript about the role of genetic ancestry in population pharmacogenetics. Some existing databases and available relevant pharmacogenetic information are analyzed to ascertain enrichment in pharmacogenetic loci, explore genetic patterns and establish useful prediction panels of genetic ancestry for global populations. A new resource named "Genetic Ancestry PhD" database to provide catalogs of AIMS and AIGs associated with drugs and related pharmacogenetics was created. Accordingly, the manuscript provides some relevant data of generalizable value that are expected to advance our current knowledge and influence thinking in the field. However, some minor issues need further consideration. See below:

Please discuss how this study compares to a previous report by Norris et al (BMC Genomics 2018,19(Suppl 8):861), particularly what concerns the identification of ancestry-enriched SNPs. Also, refer to this publication in the manuscript.

Methods, Sequencing, on page 6, lines 134-136: This is a secondary analysis of existing datasets from publicly available databases (i.e., 1KGP website); therefore, sequencing analyses were not directly performed by the authors. Please, revise this portion of the manuscript to clearly state so.

Methods, Statistical methods/ enrichment analysis of AIMS in PGx on page 9, lines 213-220: compare calculation methods versus that used by others (e.g., Norris et al., 2018).

Results, page 14, line 347: please explain the rationale for considering CYP endogenous substrates as a top major drug category of PGx.

Results, page 15, lines 353-355: please clarify why non-protein coding variants such as intronic SNVs are considered as protein-coding variants in this statement.

Discuss how the postulated genetic ancestry prediction panels derived from this study differentiate from already available commercial panels

Discussion, page 23, lines 570-575: What does it mean that a particular AI-PGx is responsible for a given drug. That is, the AI-PGx is responsible for what, drug metabolism/PK, drug response? Please, explain.

Explain whether the differential genomic structures revealed by the HD analysis in this study confirm the well-known differences of haplotype blocks across populations/ ancestries.

Reviewer #2 (Remarks to the Author):

Yang and colleagues used PCA-liked methods to re-examine population structure in 1000 Genomes Project. They then looked for genomic markers with differentiated allele frequency across populations from 1000 Genomes Project, and found an enrichment of such markers in loci associated with drug response / pharmacogenetic loci from four public databases. They further scanned for continent-specific "homozygosity disequilibrium (HD)".

The major conclusion, that pharmacogenetic loci are largely ancestry informative / differentiated in different populations, is meaningful and suggests the need for more cautious interpretation of results from pharmacogenetic associations. This echoes a similar point in Ramos et al., 2013, The Pharmacogenomics Journal. However, there're a number of analytical and interpretive concerns about the manuscript.

Major comments:

1. Re-analysis of population structure / phylogeny of 1000 Genomes Project:

A significant amount in the first half of the manuscript is about using PCA-like methods to re-examine population structure, and infer phylogeny from clustering in the public database of 1000 Genomes Project. The effort seems duplicated, and what the results serves for in this manuscript is unclear. 1000 GP has been frequently studied and ancestry components among the samples have been characterized in multiple studies / online resources (e.g. Lu and Xu, 2013; Fricot et al., 2014, Genetics; Sikora et al., 2014, Plos Genetics; Zhao et al., 2019, Forensic Science International: Genetics) http://bwlewis.github.io/1000_genomes_examples/PCA_overview.html), even the very publication of the dataset release itself (Figure 2, The 1000 Genomes Project Consortium, 2015).

It is unclear why common PCA methods would fail on the task of 1000 GP sequencings and therefore not adopted by the authors, as they have been demonstrated doable previously. If the re-analysis is about incorporating the novel algorithm (UHD-PCA) than a regular PCA, then 1) this is distracting to the main topic of the manuscript; 2) there is no systematic comparison on the performance of the new method with that from a regular PCA, to confirm a concordant outcome and a computational advantage of UHD-PCA.

Authors should consider shrinking / removing these sections, or moving them to supplementary notes if they think it's absolutely necessary to keep the results.

2. The concept of HD and technical details

"Homozygosity disequilibrium" is not a standard term to most of the readers, but only appeared in some previous publications by the same group. Therefore the authors are expected to make sure that readers understand its precise and comprehensive definition in this manuscript as well. The concept seems qualitatively equivalent to characterization of runs of homozygosity (ROH) and its deviation from neutral distribution in genomes.

The authors used LOHAS developed by them to call ROH / HD, but haven't provided any brief description on what exactly it does, parameters being used, what it outputs, in a context that readers unfamiliar with LOHAS would understand and assess the validity at least at a minimal amount.

The authors also stated that they "examine whether a median homozygosity intensity in a gene" (P10, 225) for their HD scan. Yet there is no information about how "median" HD is defined, why they chose "median" instead of "short/low", or "long/high" (if those categories exist). Analogous to ROH, different length categories are equally important as they reflect different population history (inbreeding, bottlenecks, or consanguinity etc.), and it is important to choose the "correct" category based on the tested hypothesis corresponding to each population event. Similarly, the results of HD scan to identify AIG would be biased or misleading if the choice of the length / intensity has no valid basis.

3. Homozygosity distributions by population

Based on the homozygosity disequilibrium in each continental populations, the authors found that East Asians have the largest number and widest regions of HD (P13, 304) among all global populations from 1000 GP. They attributed this to a recent bottleneck in East Asia (P19, 459).

However, East Asians didn't have most severe bottlenecks, and it has been shown that Native Americans had the most recent and severe ones, which made the overall lengths and numbers of runs of homozygosity much higher in Americans than other populations (Pemberton et al., 2012; Ceballos et al., 2018), different from what was observed here.

This major discrepancy with previous studies should be examined, especially the established population histories do not support the authors' conclusion that East Asians having the most outstanding homozygosity features. This casts doubts on the quality of properly calling runs of homozygosity in the work.

4. AIMS in each population are different. (Result- Ancestry-informative markers, figure 7 etc.)

In the manuscript, each population has different number / set of AIMS. But AIMS are by nature just markers: For example, if AIMS are found within African groups, the markers are also available in sequencings of other populations. Are some of them excluded in other groups because they're fixed in other populations? Or these AIMS are able to differentiate Africans from other groups, but potentially not able to differentiate East Asians from Europeans? The information seems unclear.

Minor comments:

1. Categorizations of genomic variants are confusing by standard and by names

The authors categorized biallelic variants based on their allele frequencies at cut-off of 1%. As 1% is usually considered ultra-rare and <1% often filtered out in a lot of QC in genetic studies, it is unclear why the threshold is chosen. Authors didn't clarify on what scale MAF is based: if allele frequency is not estimated from each population separately, but from the pooled consortium level, it is of little useful information in the subsequent analyses because of existing structures among populations. Concepts of "rare" / "common" are based on samples within HWE.

Acronyms: "RV" (rare variants) are usually variants with MAF <5% as a rule of thumb; "SNPs/RVs" reads more confusing, as it appears to be a ratio of two categories.

2. AIG are not ancestral informative

AIG, short for "ancestry informative gene" in the manuscript, were curated from scanned genes that exhibit very different homozygosity patterns in one continental population as compared to other four (P15. 368). The analysis was specific, as it set a list of genes reflecting different inbreeding strength, stochasticity etc. Naming this list of genes "ancestral informative genes" can be misleading, especially only 3 out of the ~36,000 identified genes were actually involved in discriminating continental ancestry (P16).

3. (P5) It may not be necessary to intensively describe population names and sample sizes from 1000 Genome Project. The information is extremely easy to find and summarized on their website. (P6) "Sequencing" section can be named "Quality control and filtering", since the sequencing step was done by 1000 GP.

4. (P7) The meaning of the field names such as "FX", "PK" etc. are not provided. The authors should either explain them, or remove the information, since they are not mentioned in the rest of the manuscript, nor would readers need them to help understand the work.

5. Intensive discussion of specific markers, e.g. rs11801133: These markers didn't appear in the

result section at all. It appears distracting and out of context to see pages of discussion of these variants.

6. (P24 587) "our HDA identified AIGs under differential selective pressure accross...": HD can be attributed to multiple factors other than natural selection. There is no selection strength involved in the work either.

7. Several figures are pixeled when zoomed in, and the labels / legends are not readable.

Dear Reviewers,

We are sorry for a late completion of this revision because of the COVID-19 pandemic. We are grateful to receive the reviewers' evaluations of our work (MS: COMMSBIO-19-9929-T – "Population pharmacogenomics: Enrichment of ancestry-informative markers in pharmacogenetic loci" by Hsin-Chou Yang, Chia-Wei Chen, Yu-Ting Lin, and Shih-Kai Chu). We appreciate both reviewers' helpful comments, and we have made necessary revisions accordingly. Our point-by-point responses to each of the reviewers' comments are listed below and appropriate changes have been made in the revised manuscript. In addition, we also provide a revised manuscript with track changes as a supplementary file.

Authors' Reply

To Reviewer 1

This is an interesting manuscript about the role of genetic ancestry in population pharmacogenetics. Some existing databases and available relevant pharmacogenetic information are analyzed to ascertain enrichment in pharmacogenetic loci, explore genetic patterns and establish useful prediction panels of genetic ancestry for global populations. A new resource named "Genetic Ancestry PhD" database to provide catalogs of AIMs and AIGs associated with drugs and related pharmacogenetics was created. Accordingly, the manuscript provides some relevant data of generalizable value that are expected to advance our current knowledge and influence thinking in the field. However, some minor issues need further consideration. See below:

Response:

We appreciate the summary description of our study and a good comment on our paper. We have followed the reviewer's comments to revise our manuscript accordingly.

Major comments:

1. Please discuss how this study compares to a previous report by Norris et al (BMC Genomics 2018,19(Suppl 8):861), particularly what concerns the identification of ancestry-enriched SNPs. Also, refer to this publication in the manuscript.

Response:

Thanks for providing the useful reference. Norris *et al* (2018, *BMC Genomics*, 19(Suppl 8): 861) focused on identification of "ancestry-enriched SNPs" in a mixed population. In the paper, the ancestry-enriched SNPs were defined as the SNPs whose allele frequencies in a mixed population (e.g., a modern Latin American population) were different from the expected allele frequencies in ancestral source populations (i.e., African, European, and Native American continental population groups). We identified ancestry-informative SNVs in global continents or populations rather than a single target population; in other words, we identified SNVs having a significant difference in allele frequency among the studied analysis groups (i.e., the whole-continental group and five-continent ancestry groups and also the populations in each continent) in our study. Because ancestry-enriched SNPs are also ancestry-informative loci, in the revision of this paper, we have cited Norris *et al* (2018, *BMC Genomics*, 19(Suppl 8): 861) as a publication for

identifying ancestry-informative markers in the Introduction section in Line 7 on Page 3 in the revised manuscript.

The reference cited in this reply:

E. T. Norris et al. (2018). Genetic ancestry, admixture and health determinants in Latin America. *BMC Genomics* 19 (Suppl 8): 861.

2. Methods, Sequencing, on page 6, lines 134-136: This is a secondary analysis of existing datasets from publicly available databases (i.e., 1KGP website); therefore, sequencing analyses were not directly performed by the authors. Please, revise this portion of the manuscript to clearly state so.

Response:

Thanks for the reminder. Please refer to the Quality Control and Filtering subsection in Lines -1 ~ -2 on Page 5 in the revised manuscript as follows: “All 2,504 samples were sequenced across the whole genome using either Illumina HiSeq 2000 or Illumina HiSeq 2500 by The 1000 Genomes Project.”

3. Methods, Statistical methods/ enrichment analysis of AIMS in PGx on page 9, lines 213-220: compare calculation methods versus that used by others (e.g., Norris et al., 2018).

Response:

Our enrichment analysis of ancestry-informative markers (AIMs) in pharmacogenetic loci (PGx) aimed to examine whether the proportion of AIMs in PGx was higher than the proportion of AIMs in non-PGx by using one-sided Fisher’s exact test. It’s equivalent to examine whether the proportion of AIMs in PGx was higher than the overall proportion of AIMs in human genome. The ancestry-enrichment analysis in Norris *et al* (2018, *BMC Genomics*, 19(Suppl 8): 861) aimed to examine whether SNPs in a mixed population have a higher frequency than expected based on the population ancestry profile by using a chi-squared test. Therefore, the purposes of the SNP-enrichment analysis in Norris’s study and our enrichment analysis are different although the two statistical tests (i.e., Fisher’s exact and chi-squared tests) per se are related.

4. Results, page 14, line 347: please explain the rationale for considering CYP endogenous substrates as a top major drug category of PGx.

Response:

Cytochrome P450 (CYP) is a superfamily of major enzymes involved in drug biotransformation (Maisano Delsler and Fuselli, 2013) and drug metabolism (Ingelman-Sundberg et al., 2007). In this paper, “drug category” aimed to provide a broad-sense relation of PGx and drug biotransformation and PK/PD provided in the four public PGx resources studied in this paper. The statement “... the top three major drug categories of PGx belonged to PD (N = 2,303), PK (N = 503), and CYP endogenous substrates (N = 334) (**Figure 2B**).” was made relying on the number of PGx in each drug category. All drug categories used in this study were summarized in **Figure 2A** and the number of PGx in each drug category was shown in the histogram of PGx for drug category (green bar) in the left-hand side of **Figure 2B**. Among 3,259 autosomal PGx in this study, 2,303 PGx belonged to PD, 503 PGx belonged to PK, and 334 PGx belonged to CYP endogenous

substrates. Other drug categories had the numbers of PGx smaller than 334.

To make it clearer, we add the description “Here, the drug category was defined according to the relation of PGx and drug biotransformation and PK/PD.” in the Subsection of Public PGx Resources and Functional Annotation in Lines 10 – 11 on Page 7 in the revised manuscript.

The reference cited in this reply:

M Ingelman-Sundberg, SC Sim, A Gomez, C Rodriguez-Antona. (2007). Influence of cytochrome P450 polymorphisms on drug therapies: pharmacogenetic, pharmacoeigenetic and clinical aspects. *Pharmacology and Therapeutics* 116, 496 – 526.

P Maisano Delsler & S Fuselli. (2013). Human loci involved in drug biotransformation: worldwide genetic variation, population structure, and pharmacogenetic implications. *Human Genetics* 132, 563 – 577.

5. Results, page 15, lines 353-355: please clarify why non-protein coding variants such as intronic SNVs are considered as protein-coding variants in this statement.

Response: Sorry for the typo – a pair of balanced parenthesis were lost. The description is revised as follows: “Both protein-coding variants (e.g., a missense mutation) and non-protein-coding variants (e.g., an intronic SNV) represented important mechanisms for PGx.”

6. Discuss how the postulated genetic ancestry prediction panels derived from this study differentiate from already available commercial panels.

Response:

Thanks for the suggestion. Compared to the three available commercial panels (Young et al., J. Forensic Sci, 2019), our postulated genetic ancestry prediction panel is either more accurate or parsimonious. Our panel that contains 31 AIMs and 1 AIG obtained a training accuracy of 96.6% and a testing accuracy of 95.6% in correctly assigning individuals to their continental ancestry group. The QIAGEN140-SNP Identification Multiplex panel (Young et al., J. Forensic Sci, 2019) required a larger number of SNPs (140 SNPs) and had a lower training accuracy of 95.87% and testing accuracy of 92.03% than ours. The Ion AmpliSeq HID Phenotyping Panel (Chaitanya et al., Forensic Science International Genetics, 2018) used the 24 SNP HIRISplex System and had a significantly reduced training accuracy of 81.1% and testing accuracy of 85.2% than ours. The 165 SNP Precision ID Ancestry Panel constituted by 123 SNPs from the group of M. Seldin (Kosoy et al., Human Mutation, 2009) and 55 SNPs from the group of K. Kidd (Kidd et al., the National Institute of Justice (NIJ) annual meeting, 2012) with 13 overlapping SNPs, had a higher training accuracy of 99.29% and testing accuracy of 100%, but this panel required more SNPs than ours. The results are summarized in the Discussion section in the second paragraph of Page 19.

The reference cited in this reply:

JM Young, B Martin, P. Kanokwongnuwut, A Linacre. (2019). Aust. J. Forensic Sci 7, 864 – 865.

L Chaitanya, K Breslin, S Zuñiga, L Wirken, E Pospiech, M Kukla-Bartoszek, T Sijen, P

- de Knijff, F Liu, W Branicki, M Kayser, S Walsh. (2018). *Forensic Science International Genetics* 35. 123–135.
- R Kosoy, R Nassir, C Tian, PA White, LM Butler, G Silva, R Kittles, ME Alarcon-Riquelme, PK Gregersen, JW Belmont, F M De La Vega, MF Seldin. (2009). Ancestry informative marker sets for determining continental origin and admixture proportions in common populations in America. *Human Mutation* 30: 69–78.
- KK Kidd, JR Kidd, AJ Pakstis, WC Speed. (2012). Better SNPs for better forensics: ancestry, phenotype, and family identification. Poster presented at the National Institute of Justice (NIJ) annual meeting, Arlington VA.

7. Discussion, page 23, lines 570-575: What does it mean that a particular AI-PGx is responsible for a given drug. That is, the AI-PGx is responsible for what, drug metabolism/PK, drug response? Please, explain.

Response:

When we described “a particular AI-PGx is responsible for a given drug”, it indicates that AI-PGx was reported to be associated or responsible for drug PK/PD. For illustration, we use rs4646440 on *CYP3A4* as an example. Our analysis showed rs4646440 was an AIM (ancestry-informative marker). Moreover, PharmGKB reported that rs4646440 was related to drug methadone (<https://www.pharmgkb.org/variant/PA166157393/variantAnnotation>) as follows: “Allele A is associated with increased severity of side effects when treated with methadone in people with Heroin Dependence as compared to allele G” and “Allele A is associated with increased severity of opioid withdrawal symptoms when treated with methadone in people with Heroin Dependence as compared to allele G.” Based on the results, we described “Rs4646440 was an AI-PGx responsible for the drug methadone” in the manuscript. We did not describe the detailed drug category in the manuscript because the information can be extracted from our genetic ancestry pharmacogenomic database “Genetic Ancestry PhD”. We appreciate the reviewer’s reminder. We revise the description “a particular AI-PGx is responsible for a given drug” by “a particular AI-PGx is responsible for the PK/PD of a given drug”. For example, in the revised manuscript, we add an illustration from Line -2 on Page 22 to Line 2 on Page 23: “Rs4646440 was an AI-PGx responsible for the PK/PD of the drug methadone. The details of the drug categories triggered by or associated with the AI-PGx can be accessed from our genetic ancestry pharmacogenomic database “Genetic Ancestry PhD”.”

8. Explain whether the differential genomic structures revealed by the HD analysis in this study confirm the well-known differences of haplotype blocks across populations/ancestries.

Response:

Homozygosity disequilibrium (HD) can be caused by different mechanisms as mentioned in the Introduction section of this paper. HD is related but not equivalent to haplotype blocks that show high levels of linkage disequilibrium (LD) and are separated from one another by a number of recombination events. For illustration, a toy example of three SNPs is given as follows: if the two major haplotype configurations are {A/B/C, A/B/C} and {A/b/c, A/b/c} and each of them has a frequency of 45%. In this region, it has high homozygosity but low LD. We did not perform a systematic comparison of the genomic

regions in HD and haplotype blocks across populations/ancestries because it is out of this paper's scope. We leave the detailed investigation as future work.

To Reviewer 2

Yang and colleagues used PCA-liked methods to re-examine population structure in 1000 Genomes Project. They then looked for genomic markers with differentiated allele frequency across populations from 1000 Genomes Project, and found an enrichment of such markers in loci associated with drug response / pharmacogenetic loci from four public databases. They further scanned for continent-specific "homozygosity disequilibrium (HD)". The major conclusion, that pharmacogenetic loci are largely ancestry informative / differentiated in different populations, is meaningful and suggests the need for more cautious interpretation of results from pharmacogenetic associations. This echoes a similar point in Ramos et al., 2013, The Pharmacogenomics Journal. However, there're a number of analytical and interpretive concerns about the manuscript.

Response: We thank this reviewer for the useful comments. We have answered and followed the reviewer's comments to revise our manuscript accordingly.

Major comments:

1. Re-analyzation of population structure / phylogeny of 1000 Genomes Project:

A significant amount in the first half of the manuscript is about using PCA-like methods to re-examine population structure, and infer phylogeny from clustering in the public database of 1000 Genomes Project. The effort seems duplicated, and what the results serves for in this manuscript is unclear. 1000 GP has been frequently studied and ancestry components among the samples have been characterized in multiple studies / online resources (e.g. Lu and Xu, 2013; Frichot et al., 2014, Genetics; Sikora et al., 2014, Plos Genetics; Zhao et al., 2019, Forensic Science International: Genetics) http://bwlewis.github.io/1000_genomes_examples/PCA_overview.html); even the very publication of the dataset release itself (Figure 2, The 1000 Genomes Project Consortium, 2015). It is unclear why common PCA methods would fail on the task of 1000 GP sequencings and therefore not adopted by the authors, as they have been demonstrated doable previously. If the re-analysis is about incorporating the novel algorithm (UHD-PCA) than a regular PCA, then 1) this is distracting to the main topic of the manuscript; 2) there is no systematic comparison on the performance of the new method with that from a regular PCA, to confirm a concordant outcome and a computational advantage of UHD-PCA. Authors should consider shrinking / removing these sections, or moving them to supplementary notes if they think it's absolutely necessary to keep the results.

Response:

We follow the reviewer's suggestion to move the detailed results of population structure to **Supplementary Text S1**.

Compared to the original PCA algorithm, the proposed UHD-PCA provides an efficient alternative in terms of computational memory and the results are the same (without a loss of estimation accuracy). It is especially suitable for the large data set in a

whole-genome sequencing data analysis. We add this description to the bottom of **Appendix A**.

2. The concept of HD and technical details: “Homozygosity disequilibrium” is not a standard term to most of the readers, but only appeared in some previous publications by the same group. Therefore the authors are expected to make sure that readers understand its precise and comprehensive definition in this manuscript as well. The concept seems qualitatively equivalent to characterization of runs of homozygosity (ROH) and its deviation from neutral distribution in genomes. The authors used LOHAS developed by them to call ROH / HD, but haven’t provided any brief description on what exactly it does, parameters being used, what it outputs, in a context that readers unfamiliar with LOHAS would understand and assess the validity at least at a minimal amount. The authors also stated that they “examine whether a median homozygosity intensity in a gene” (P10, 225) for their HD scan. Yet there is no information about how “median” HD is defined, why they chose “median” instead of “short/low”, or “long/high” (if those categories exist). Analogous to ROH, different length categories are equally important as they reflect different population history (inbreeding, bottlenecks, or consanguinity etc.), and it is important to choose the “correct” category based on the tested hypothesis corresponding to each population event. Similarly, the results of HD scan to identify AIG would be biased or misleading if the choice of the length / intensity has no valid basis.

Response:

We elaborate more about how we detected homozygosity disequilibrium by using LOHAS in the Homozygosity Disequilibrium Analysis subsection in the Statistical Methods section on Page 8 in the revised manuscript as follows:

“Homozygosity disequilibrium (HD), originally coined by Yang et al (H. C. Yang, Chang, Liang, Lin, & Wang, 2012), is defined by a non-random pattern of sizable run of homozygosity where its homozygosity intensity exceeds the value under equilibrium in the human genome. Homozygosity intensity can be estimated based on SNV genotypes and its estimate ranges between 0 and 1. A higher value of homozygosity intensity indicates a higher homozygosity in a genomic region. The procedures of our homozygosity disequilibrium analysis (HDA) are described as follows. First, genome-wide profile of homozygosity intensity for every individual was calculated based on genotype data under a double-weight local polynomial model by using LOHAS version 2.3 (H.-C. Yang & Lin, 2015; H. C. Yang, Chang, Huggins, Chen, & Mullighan, 2011; H. C. Yang & Chen, 2018; H. C. Yang & Lin, 2016). The double weights were composed by a cubic kernel weight for considering a local smoothing property and a locus weight with a threshold of minor allele frequency of 0.001 for adjusting for low-informativeness of RVs. The procedure was applied to each gene region to estimate homozygosity intensity of a gene for each individual. Second, in each gene, the homozygosity intensities for each population or for all populations within each of the six analysis groups (refer to the Samples subsection) were summarized by taking an average (median and mean) homozygosity intensity of all individuals in the population or analysis group, respectively. Third, for all gene regions and the gene regions that contain PGx, the genomic distributions of the number and length of the regions under HD (i.e., an average homozygosity intensity in a gene region is of >0.9) in each population and continental ancestry group were summarized in violin plots.”

We agree with the reviewer's point that different length categories may reflect different population history when run of homozygosity is investigated. However, in order to consider biological functions, we conducted a "gene-centric" homozygosity analysis in this study. The length of a gene is fixed. Importantly, homozygosity disequilibrium is related to but not equal to run of homozygosity. In addition to the length, we are also interested in the differential magnitude of homozygosity intensities in ancestry groups. In the revised manuscript, we give a short discussion on the size of the genes under homozygosity disequilibrium in the Discussion section in Lines 2 – 8 on Page 25 as follows: "Different lengths of homozygosity regions may reflect different population history. A short segment of homozygosity in tens of KB may be formed by a pair of ancient haplotypes that contribute to local LD patterns (Pemberton et al., 2012; Ceballos et al., 2018); however, it is not always the case. CEU and GBR did not exhibit a striking difference in LD patterns compared to other European- and non-European-ancestry groups in *LCT* and *MCM6*. Previous study demonstrated that *LCT* and *MCM6* acted as a selective pressure in the European-ancestry group¹⁰⁰."

The references cited in this reply:

- Yang, H. C., Chang, L. C., Liang, Y. J., Lin, C. H., & Wang, P. L. (2012). A genome-wide homozygosity association study identifies runs of homozygosity associated with rheumatoid arthritis in the human major histocompatibility complex. *PLoS ONE*, 7(4), e34840. doi:10.1371/journal.pone.0034840PONE-D-11-09354 [pii]
- Yang, H.-C., & Lin, Y.-T. (2015). *Homozygosity disequilibrium in the human genome*. Paper presented at the The Conference of HGM 2015, 007., Kuala Lumpur, Malaysia.
- Yang, H. C., Chang, L. C., Huggins, R. M., Chen, C. H., & Mullighan, C. G. (2011). LOHAS: loss-of-heterozygosity analysis suite. *Genetic Epidemiology*, 35(4), 247-260. doi:10.1002/gepi.20573
- Yang, H. C., Chang, L. C., Liang, Y. J., Lin, C. H., & Wang, P. L. (2012). A genome-wide homozygosity association study identifies runs of homozygosity associated with rheumatoid arthritis in the human major histocompatibility complex. *PLoS ONE*, 7(4), e34840. doi:10.1371/journal.pone.0034840PONE-D-11-09354 [pii]
- Yang, H. C., & Chen, C. W. (2018). Homozygosity disequilibrium associated with treatment response and its methylation regulation. *BMC Proc*, 12(Suppl 9), 45. doi:10.1186/s12919-018-0150-9
- Yang, H. C., & Lin, Y. T. (2016). Homozygosity disequilibrium and its gene regulation. *BMC Proceedings*, 10(Suppl 7), 159-163. doi:10.1186/s12919-016-0023-z23 [pii]
- Pemberton, T. J. et al. (2012). Genomic patterns of homozygosity in worldwide human populations. *American Journal of Human Genetics* 91, 275-292.
- Ceballos, F.C., Joshi, P.K., Clark, D.W., Ramsay, M. & Wilson, J.F. Runs of homozygosity: windows into population history and trait architecture. *Nature Reviews Genetics* 19, 220 (2018).

3. Homozygosity distributions by population: Based on the homozygosity disequilibrium in each continental populations, the authors found that East Asians have the largest number and widest regions of HD (P13, 304) among all global populations from 1000 GP. They attributed this to a recent bottleneck in East Asia (P19, 459). However, East

Asians didn't have most severe bottlenecks, and it has been shown that Native Americans had the most recent and severe ones, which made the overall lengths and numbers of runs of homozygosity much higher in Americans than other populations (Pemberton et al., 2012; Ceballos et al., 2018), different from what was observed here. This major discrepancy with previous studies should be examined, especially the established population histories do not support the authors' conclusion that East Asians having the most outstanding homozygosity features. This casts doubts on the quality of properly calling runs of homozygosity in the work.

Response:

The discrepancy raised by this reviewer actually came from that the Americans-ancestry populations in The 1000 Genomes Project studied in this paper differed from the populations in the Human Genome Diversity Panel (HGDP) data (Li et al., 2008, *Science*; Pickrell et al, 2009, *Genome Research*). The reason is elaborated in details from Line -2 on Page 17 to Line -5 on Page 18 follows: “In this study, we found that East Asians carried the largest number and widest regions of homozygosity disequilibrium in The 1000 Genomes Project. This finding is not in conflict with the previous finding (Pemberton et al., 2012; Ceballos et al., 2018) that Native Americans had the most recent and severe bottlenecks, which made the overall lengths and numbers of runs of homozygosity much higher in Americans than other populations. The Americans-ancestry populations in The 1000 Genomes Project studied in this paper differ from the populations in the Human Genome Diversity Panel (HGDP) data. In HGDP, the Americans-ancestry populations (Maya, Pima, Colombian, Karitiana, and Surul) are Native Americans. These Native Americans populations had undergone the recent and severe bottlenecks and exhibited the much higher lengths of runs of homozygosity than other populations (Refer to **Figure 3** in Pemberton et al. (*AJHG*, 2012)). In The 1000 Genomes Project, the Americans-ancestry populations (MXL, PUR, CLM, and PEL) are admixed Americans. This ancestry admixture reflects in the large variability of the lengths and numbers of homozygosity disequilibrium (Refer to **Figure 6** in this paper). MXL were included in both of The 1000 Genomes Project and the International Haplotype Map Project III (The International HapMap 3 Consortium, 2010, *Nature*). This admixed Americans population did not show higher lengths of runs of homozygosity compared to East Asians (Refer to **Figure 3** in Pemberton et al. (*AJHG*, 2012)), and this result is consistent to our finding. The CLM participants in The 1000 Genomes Project were Colombians with admixed ancestry recruited from the second-largest city in Colombia and they differed from the Colombian participants with the Native Americans ancestry in the HGDP. As expected, the Colombian participants in the HGDP exhibited much higher lengths of runs of homozygosity compared to the CLM in The 1000 Genomes Project (Refer to **Figure 3** in Pemberton et al. (*AJHG*, 2012)).” Our reply to this comment is provided in the Discussion section from Line -2 on Page 17 to Line 21 on Page 18 in the revised manuscript.

The references cited in this reply:

- Li, J. Z. et al. (2008) Worldwide human relationships inferred from genome-wide patterns of variation. *Science* 319, 1100-1104.
- Pickrell, J. K. et al. Signals of recent positive selection in a worldwide sample of human populations (2009). *Genome Research* 19, 826-837.

Pemberton, T. J. et al. (2012). Genomic patterns of homozygosity in worldwide human populations. *American Journal of Human Genetics* 91, 275-292.

Ceballos, F.C., Joshi, P.K., Clark, D.W., Ramsay, M. & Wilson, J.F. Runs of homozygosity: windows into population history and trait architecture. *Nature Reviews Genetics* 19, 220 (2018).

The International HapMap 3 Consortium. (2010) Integrating common and rare genetic variation in diverse human populations. *Nature* 467, 52-58.

4. AIMS in each population are different. (Result- Ancestry-informative markers, figure 7 etc.) In the manuscript, each population has different number / set of AIMS. But AIMS are by nature just markers: For example, if AIMS are found within African groups, the markers are also available in sequencings of other populations. Are some of them excluded in other groups because they're fixed in other populations? Or these AIMS are able to differentiate Africans from other groups, but potentially not able to differentiate East Asians from Europeans? The information seems unclear.

Response:

In this paper, AIMS are the markers that are able to differentiate ancestry populations within an analysis group (please refer to the Materials and Methods section for the six analysis groups). Some AIMS are able to differentiate populations in the Asian-ancestry group but cannot differentiate populations in the European-Ancestry group. **Figure 7** summarizes the number of AIMS that can differentiate populations in an analysis group (not the number of AIMS within a population). For example, in **Figure 7(A)**, it shows that 2,725,350 AIMS that can differentiate populations in AFR were identified, and these markers cannot differentiate populations in AMR, EAS, SAS, and EUR individually.

Minor comments:

1. Categorizations of genomic variants are confusing by standard and by names: The authors categorized biallelic variants based on their allele frequencies at cut-off of 1%. As 1% is usually considered ultra-rare and <1% often filtered out in a lot of QC in genetic studies, it is unclear why the threshold is chosen. Authors didn't clarify on what scale MAF is based: if allele frequency is not estimated from each population separately, but from the pooled consortium level, it is of little useful information in the subsequent analyses because of existing structures among populations. Concepts of "rare" /"common" are based on samples within HWE. Acronyms: "RV" (rare variants) are usually variants with MAF <5% as a rule of thumb; "SNPs/RVs" reads more confusing, as it appears to be a ratio of two categories.

Response:

About the definition of a "polymorphism" (e.g., single nucleotide polymorphism; SNP), the threshold of 1% in minor allele frequency has been adopted for many years. In the official website of National Human Genome Research Institute (<https://www.genome.gov/genetics-glossary/Polymorphism>), "Polymorphism, by strict definitions which hardly anybody pays attention to anymore, is a place in the DNA sequence where there is variation, and the less common variant is present in at least one percent of the people of who you test." In Wikipedia (https://en.wikipedia.org/wiki/Single-nucleotide_polymorphism), "A single-nucleotide

polymorphism (SNP; /snp/; plural /snps/) is a substitution of a single nucleotide at a specific position in the genome, that is present in a sufficiently large fraction of the population (e.g. 1% or more).” We understand that a threshold of 0.05 was also adopted in some papers. However, a threshold of 0.01 has been commonly used in many papers.

In a disease gene mapping study, single nucleotide variations (SNVs) with a minor allele frequency of <0.01 were removed. This is because of the “common disease, common variant” (CDCV) assumption especially for genome-wide association studies based on SNP microarrays that mainly interrogate common SNP probes on chips. However, the CDCV assumption may not hold and has led to a serious issue of missing heritability (Eichler et al., 2010; Maher, 2008; Manolio et al., 2009). Now, the “common disease, rare variant” (CDRV) scenario has been broadly realized and accepted (Schork, Murray, Frazer, & Topol, 2009). Moreover, the next-generation sequencing (NGS) generates a large number of SNVs including common SNPs and rare variants (RVs). Many RV analysis methods (Asimit & Zeggini, 2010; Povysil et al., 2019) have been developed based on NGS data, and SNVs with a minor allele frequency of <0.01 must be included in the analysis because they provide information for rare and common diseases and for population genetics (Nagasaki et al., 2015; The 1000 Genomes Project Consortium, 2015). In this study, allele frequency was estimated from each study group.

About RV, we really hesitate to call the SNVs with MAF of <5% as RVs because this will cause conflicts between this paper and many other papers. To avoid a potential confusion, we only replace SNPs/RVs with “SNPs and/or RVs”.

The references cited in this reply:

- Asimit, J., & Zeggini, E. (2010). Rare variant association analysis methods for complex traits. *Annual Review of Genetics*, *44*, 293-308. doi:10.1146/annurev-genet-102209-163421
- Eichler, E. E., Flint, J., Gibson, G., Kong, A., Leal, S. M., Moore, J. H., & Nadeau, J. H. (2010). Missing heritability and strategies for finding the underlying causes of complex disease. *Nature Reviews Genetics*, *11*(6), 446-450. doi:nrg2809 [pii] 10.1038/nrg2809
- Maher, B. (2008). Personal genomes: The case of the missing heritability. *Nature*, *456*(7218), 18-21. doi:10.1038/456018a
- Manolio, T. A., Collins, F. S., Cox, N. J., Goldstein, D. B., Hindorff, L. A., Hunter, D. J., . . . Visscher, P. M. (2009). Finding the missing heritability of complex diseases. *Nature*, *461*(7265), 747-753. doi:nature08494 [pii] 10.1038/nature08494
- Nagasaki, M., Yasuda, J., Katsuoka, F., Nariyai, N., Kojima, K., Kawai, Y., . . . Yamamoto, M. (2015). Rare variant discovery by deep whole-genome sequencing of 1,070 Japanese individuals. *Nat Commun*, *6*, 8018. doi:10.1038/ncomms9018
- Povysil, G., Petrovski, S., Hostyk, J., Aggarwal, V., Allen, A. S., & Goldstein, D. B. (2019). Rare-variant collapsing analyses for complex traits: guidelines and applications. *Nat Rev Genet*. doi:10.1038/s41576-019-0177-4
- Schork, N. J., Murray, S. S., Frazer, K. A., & Topol, E. J. (2009). Common vs. rare allele hypotheses for complex diseases. *Current Opinion in Genetics & Development*, *19*(3), 212-219. doi:10.1016/j.gde.2009.04.010

The 1000 Genomes Project Consortium. (2015). A global reference for human genetic variation. *Nature*, 526(7571), 68-74. doi:10.1038/nature15393
nature15393 [pii]

2. AIG are not ancestral informative: AIG, short for “ancestry informative gene” in the manuscript, were curated from scanned genes that exhibit very different homozygosity patterns in one continental population as compared to other four (P15. 368). The analysis was specific, as it set a list of genes reflecting different inbreeding strength, stochasticity etc. Naming this list of genes “ancestral informative genes” can be misleading, especially only 3 out of the ~36,000 identified genes were actually involved in discriminating continental ancestry (P16).

Response:

Figure S1 revealed that homozygosity intensity is able to differentiate ancestry groups and the panel “Whole-Continent” in **Figure 8** revealed very significant differences in homozygosity patterns across five ancestry groups (AFR, AMR, EAS, EUR, and SAS). In the section of Ancestry Informative Gene, “Manhattan plots from a genome-wide homozygosity association study are displayed (**Figure 8**). Among 37,049 gene regions, we found 99.360% of AIGs (N = 36,812) under HD among continents. In specific continent, we found 2.904% (N = 1,076), 34.562% (N = 12,805), 4.802% (N = 1,779), 2.440% (N = 904), and 0.432% (N = 160) AIGs under HD in AFR, AMR, EAS, EUR, and SAS, respectively.” The number of AIGs is not low. These results point out AIG are truly ancestral informative (although the information may be not richer than genotype data). Because ancestry information explained by SNV-based AIGs and homozygosity-based AIGs are overlapped and a parsimonious principle of genetic ancestry prediction model was applied, only few AIGs were included in the final genetic ancestry prediction model.

73. (P5) It may not be necessary to intensively describe population names and sample sizes from 1000 Genome Project. The information is extremely easy to find and summarized on their website. (P6) “Sequencing” section can be named “Quality control and filtering”, since the sequencing step was done by 1000 GP.

Response:

Because the population names and their abbreviations were mentioned frequently in this paper, we remove the description of population names and samples from the Samples subsection in the Materials and Methods section but move to the caption of **Figure 1**. We have followed the reviewer’s suggestion to rename “Sequencing” section as “Quality control and filtering” in Line -3 on Page 5 in the revised manuscript.

4. (P7) The meaning of the field names such as “FX”, “PK” etc. are not provided. The authors should either explain them, or remove the information, since they are not mentioned in the rest of the manuscript, nor would readers need them to help understand the work.

Response:

Thank you for the reminder. The terms and their corresponding abbreviations are added on Page 3 in the revised manuscript – pharmacodynamics (PD), pharmacokinetics (PK), drug

response (DR), pharmacogenomic effect (FX), adverse drug reaction (ADR), and drug's effective dose (DED).

5. Intensive discussion of specific markers, e.g. rs1801133: These markers didn't appear in the result section at all. It appears distracting and out of context to see pages of discussion of these variants.

Response:

This study identified AIMS, AIGs, and AI-PGx in the global continents and populations and archived the results in our genetic ancestry pharmacogenomic database "Genetic Ancestry PhD" (http://hcyang.stat.sinica.edu.tw/databases/genetic_ancestry_phd/). Aiming to provide the readers a better understanding about our findings, a few important examples and discussions are given in the Discussion section.

6. (P24 587) "our HDA identified AIGs under differential selective pressure across...": HD can be attributed to multiple factors other than natural selection. There is no selection strength involved in the work either.

Response:

We rephrase the description and provide several references on Lines -7 ~ -9 on Page 23 as follows: "In addition, our HDA identified several AIGs with reported evidence of natural selection such as the *CYP3A* family (Thompson et al., *AJHG*, 2004; Chen et al., *Environmental Health Perspectives*, 2009), *LCT* (Bersaglieri et al., *AJHG*, 2004; Tishkoff et al., *Nature Genetics*, 2007), and *MCM6* (Tishkoff et al., *Nature Genetics*, 2007; Enattah et al., *AJHG*, 2008)."

The references cited in this reply:

Thompson et al. (2004) *CYP3A* variation and the evolution of salt-sensitivity variants. *AJHG* **75**, 1059 – 1069.

Chen et al. (2009) Molecular population genetics of human *CYP3A* locus: Signatures of positive selection and implications for evolutionary environmental medicine. *Environmental Health Perspectives* **117**, 1041 – 1048.

Bersaglieri et al. (2004) Genetic signature of strong recent positive selection at the lactase gene. *AJHG* **74**, 1111 – 1120.

Tishkoff et al. (2007). Convergent adaptation of human lactase persistence in Africa and Europe. *Nat. Genet.* **39**, 31 – 40.

Enattah et al. (2008). Independent introduction of two lactase-persistence alleles into human populations reflects different history of adaptation to milk culture. *AJHG* **82**, 57 – 72.

7. Several figures are pixelated when zoomed in, and the labels / legends are not readable.

Response:

We apologize for that and have increased the font size of the labels / legends. About that several figures are pixelated when zoomed in, it may be caused by the file conversion or manuscript combination in the manuscript submission system. We follow the guide of Figures for Publication in *Communications Biology* to prepare all figures, and make sure the labels and legends are readable.

REVIEWERS' COMMENTS:

Reviewer #1 (Remarks to the Author):

No further comments.

Reviewer #4 (Remarks to the Author):

The manuscript rebuttal and revisions have appropriately addressed all comments/suggestions/concerns initially raised by Reviewer #2. I have no additional comments for this manuscript.